# The novel role of Kallistatin in linking metabolic syndromes and cognitive memory deterioration by inducing amyloid-β plaques accumulation and tau protein hyperphosphorylation

Weiwei Qi[1,2†], Yanlan Long[3†], Ziming Li[4†], Zhen Zhao[1], Jinhui Shi[1], Wanting Xie[1], Laijian Wang[4], Yandan Tan[1], Ti Zhou[1], Minting Liang[1], Ping Jiang[5*], Bin Jiang[4*], Xia Yang[1,6*], Guoquan Gao[1,7,8,9*]

[1]Department of Biochemistry and Molecular Biology, Zhongshan School of Medicine, Sun Yat-sen University, Guangzhou, China; [2]Advanced Medical Technology Center, The First Affiliated Hospital, Zhongshan School of Medicine, Sun Yat-sen University, Guangzhou, China; [3]Guangdong Key Laboratory of Nanomedicine, CAS-HK Joint Lab of Biomaterials, Shenzhen Institute of Advanced Technology (SIAT), Chinese Academy of Sciences (CAS), Shenzhen, China; [4]Guangdong Province Key Laboratory of Brain Function and Disease, School of Medicine, Sun Yat-sen University, Shenzhen, China; [5]Department of Clinical Medical Laboratory, Guangzhou First People Hospital, School of Medicine, South China University of Technology, Guangzhou, China; [6]China Key Laboratory of Tropical Disease Control (Sun Yat-sen University), Ministry of Education, Guangzhou, China; [7]Guangdong Engineering & Technology Research Center for Gene Manipulation and Biomacromolecular Products (Sun Yat-sen University), Guangzhou, China; [8]Guangdong Province Key Laboratory of Brain Function and Disease, Zhongshan School of Medicine, Sun Yat-sen University, Guangzhou, China; [9]Guangdong Provincial Key Laboratory of Diabetology & Guangzhou Municipal Key Laboratory of Mechanistic and Translational Obesity Research, Medical Center for Comprehensive Weight Control, The Third Affiliated Hospital of Sun Yat-sen University Guangzhou, Guangdong, China

*For correspondence:
jiangp45@mail2.sysu.edu.cn (PJ);
jiangb3@mail.sysu.edu.cn (BJ);
yangxia@mail.sysu.edu.cn (XY);
gaogq@mail.sysu.edu.cn (GG)

†These authors contributed equally to this work

Competing interest: The authors declare that no competing interests exist.

## eLife Assessment

This **important** study identified a molecular mechanism linking diabetes to AD risk and the data presented are **convincing**. The authors investigated the role of kallistatin in metabolic abnormalities associated with AD and identified that Kallistatin is a key player that mediates Aβ accumulation and tau hyperphosphorylation in AD. This manuscript provides novel insights into the pathogenesis of AD, indicating that the hypolipidemic drug fenofibrate attenuates AD-like pathology in Kallistatin transgenic mice.

**Abstract** Accumulation of amyloid-β (Aβ) peptides and hyperphosphorylated tau proteins in the hippocampus triggers cognitive memory decline in Alzheimer's disease (AD). The incidence and mortality of sporadic AD were tightly associated with diabetes and hyperlipidemia, while the exact linked molecular mechanism is uncertain. Here, the present investigation identified significantly

elevated serum Kallistatin levels in AD patients concomitant with hyperglycemia and hypertriglyceridemia, suggesting potential crosstalk between neuroendocrine regulation and metabolic dysregulation in AD pathophysiology. In addition, the constructed Kallistatin-transgenic (KAL-TG) mice defined its cognitive memory impairment phenotype and lower long-term potentiation in hippocampal CA1 neurons accompanied by increased Aβ deposition and tau phosphorylation. Mechanistically, Kallistatin could directly bind to the Notch1 receptor and thereby upregulate BACE1 expression by inhibiting PPARγ signaling, resulting in Aβ cleavage and production. Besides, Kallistatin could promote the phosphorylation of tau by activating GSK-3β. Fenofibrate, a hypolipidemic drug, could alleviate cognitive memory impairment by downregulating Aβ and tau phosphorylation of KAL-TG mice. Collectively, the experiments clarified a novel mechanism for Aβ accumulation and tau protein hyperphosphorylation regulation by Kallistatin, which might play a crucial role in linking metabolic syndromes and cognitive memory deterioration, and suggested that fenofibrate might have the potential for treating metabolism-related AD.

## Introduction

Alzheimer's disease (AD), the most prevalent irreversible neurodegenerative disorder associated with dementia in elderly individuals, is marked by a gradual decline in cognitive memory. Pathologically, AD is identified by the presence of extracellular amyloid-β (Aβ) plaques and intracellular neurofibrillary tangles (NFTs) (*Grundke-Iqbal et al., 1986*; *Haass and Selkoe, 2007*; *Holtzman et al., 2011*). The Aβ cascade and tau protein hyperphosphorylation are the two primary hypotheses concerning AD. According to the Aβ cascade hypothesis, the excessive production of Aβ disrupts normal cellular functions, leading to synaptic dysfunction, neurodegeneration, tau hyperphosphorylation, and neuroinflammation, which ultimately result in memory impairment in individuals with AD and dementia (*Leng and Edison, 2021*; *De Strooper and Karran, 2016*). Aβ peptides are derived from the sequential cleavage of amyloid precursor protein (APP) by β-secretase (β-site APP cleaving enzyme 1, BACE1) and γ-secretase, thus making this cleavage process significant in AD pathology (*Tomita, 2014*; *LaFerla et al., 2007*). BACE1 is considered a highly promising therapeutic target. Several potent BACE1 inhibitors have progressed to advanced stages in clinical trials, emphasizing the role of BACE1 in Aβ production (*Das and Yan, 2019*; *Crunkhorn, 2017*; *Yan and Vassar, 2014*). Tau, a microtubule-associated protein, naturally occurs in axons and regulates microtubule dynamics and axonal transport (*Wang and Mandelkow, 2016*). In AD, tau undergoes a multistep transformation from a natively unfolded monomer to large aggregated forms, such as NFTs, another defining feature of AD (*Hausrat et al., 2022*; *Braak et al., 1994*). Glycogen synthase kinase-3 (GSK3) is a key kinase involved in the initial steps of tau phosphorylation, with Wnt signaling being crucial in activating GSK-3β and GSK-3β-mediated tau phosphorylation (*Boonen et al., 2009*). The physiological mechanisms underlying their interaction remain poorly understood.

There is a close relationship between metabolic disorders and cognitive impairment across the AD spectrum (*Biessels and Despa, 2018*; *Li et al., 2017*). Nearly 95% of AD patients are categorized as sporadic patients, whose increasing incidence and mortality are strongly associated with type 2 diabetes mellitus (T2DM), obesity, and hyperlipidemia (*Cardoso et al., 2017*; *Schipper, 2011*; *Folch et al., 2015*). About 37% of comorbidities between AD and diabetes have been reported in the Alzheimer's Association Report (*Alzheimer's Association, 2016*; *Baglietto-Vargas et al., 2016*). As a result of the strong association and shared mechanism between AD and T2DM, AD has been termed 'type 3 diabetes' by some researchers (*Kandimalla et al., 2017*; *Zhang et al., 2018*; *Steen et al., 2005*; *Candasamy et al., 2020*). Several studies have demonstrated that diabetes confers a 1.6-fold increased risk of developing dementia (*Koekkoek et al., 2015*; *Hildreth et al., 2012*). Similarly, central obesity and high body mass index during middle age are associated with about a 3.5-fold increased risk of dementia later in life (*Anjum et al., 2018*). Therefore, controlling blood glucose and lipids is expected to be a strategy for preventing or moderating cognitive decline during aging. Nevertheless, the exact link and key associated regulators between metabolic abnormalities and AD are still unclear.

Kallistatin is a serine proteinase inhibitor previously identified as a tissue kallikrein-binding protein (*Ma et al., 2018*). It is produced predominantly in the liver and is widely expressed in body tissues, where it has antiangiogenic, antifibrotic, antioxidative stress, and antitumor effects (*Yang et al., 2020*;

*Ma et al., 2017*). Furthermore, Kallistatin was found to be increased in patients with obesity, predia-betes, and diabetes (*El-Asrar et al., 2015*; *Gateva et al., 2017*; *Campbell et al., 2010*). The concentration of Kallistatin was positively correlated with the triglyceride glucose index (*Nowicki et al., 2021*), which was proven to be an independent risk factor for dementia (*Hong et al., 2021*). In addition, evidence from our previous study revealed that the concentration of serum Kallistatin in T2DM patients was significantly increased and further revealed that Kallistatin suppressed wound healing in T2DM patients by promoting local inflammation, which suggested that Kallistatin plays a critical role in the progression of T2DM (*Feng et al., 2019*). Furthermore, it was revealed that Kallistatin can cause memory and cognitive dysfunction by disrupting the glutamate–glutamine cycle (*Long et al., 2024*).

To investigate the pathophysiological interplay between T2DM, AD, and Kallistatin, a transgenic murine model overexpressing Kallistatin (KAL-TG) was developed, aiming to elucidate the mechanistic basis of Kallistatin-induced cognitive decline through modulation of Aβ generation. Taken together, these experimental findings suggest that a novel regulatory mechanism of Aβ production and tau protein hyperphosphorylation by Kallistatin is involved in the progression of metabolic abnormality-related AD.

## Results

### Kallistatin increases in AD patients and AD model mice

To explore the relevance of AD in T2DM, a GAD disease enrichment analysis was initially conducted on differentially expressed genes in neurons of T2DM patients and normal controls, revealing a close relationship between AD and T2DM (GSE161355) (*Figure 1—figure supplement 1A*). Additionally, PFAM analysis using the DAVID database identified enrichment of the Serpin family protein domain (*Figure 1—figure supplement 1B*) (https://david.ncifcrf.gov/). Kallistatin (serpin family a member 4) was elevated in the serum of T2DM patients and was associated with an adverse prognosis of diabetes complications (*Feng et al., 2019*). Serum samples were collected from 11 dementia patients at Sun Yat-sen Memorial Hospital, and Kallistatin concentrations were observed to be significantly elevated compared to those in healthy control subjects (*Long et al., 2024*). Fifty-six AD patients and 61 healthy controls were enrolled from four hospitals in Guangdong Province to further investigate the potential relevance of Kallistatin and AD. The clinical and biochemical characteristics of the participants are provided in *Supplementary file 1* (Tables S1 and S2). In addition, the serum Kallistatin (12.78 ± 2.80 μg/ml) content in patients with AD was greater than that in normal controls (9.78 ± 1.93 μg/ml) (*Figure 1A*). Similarly, fasting blood glucose and triglyceride (TG) levels were greater in AD patients than in healthy controls (*Figure 1B*). Further according to diabetes status, all the AD patients were grouped and found that the Kallistatin (13.79 ± 3.05 μg/ml) and TG contents were further elevated in AD patients with diabetes (*Figure 1C, D*). Similarly, Kallistatin expression was increased in the hippocampus of the AD model mouse SAMP8 compared with that in the control mouse SAMR1 (*Figure 1—figure supplement 1C, D*). Taken together, these results indicate that the Kallistatin concentration is increased in metabolic abnormality-related AD patients.

### Kallistatin impairs cognitive memory in mice

The aforementioned experimental data indicated elevated Kallistatin levels in both AD patients and AD model mice. KAL-TG mice were subsequently generated, and their behavioral phenotypes were evaluated using the Morris water maze (MWM) and Y-maze tests. Notably, the latency to escape the platform was longer, and the number of platform crossings, percentage of time spent, and spontaneous alternation were significantly lower in KAL-TG mice than in age-matched WT mice (*Figure 1E–I*). Furthermore, long-term potentiation (LTP) was measured using whole-cell voltage-clamp recordings of CA1 neurons in acute hippocampal slices from KAL-TG and WT mice to assess changes in hippocampal synapses. The LTP in KAL-TG mice was significantly reduced compared to that in WT mice (*Figure 1J*). Collectively, these findings suggest that Kallistatin could impair cognitive memory in mice.

### Kallistatin promotes Aβ deposition and tau phosphorylation

In order to observe the pathological phenomena of KAL-TG mice, Aβ deposition and tau phosphorylation in hippocampal tissues from experimental mice were assessed through immunohistochemistry (IHC) and western blotting. Predictably, the plaque density and tau phosphorylation in KAL-TG mice

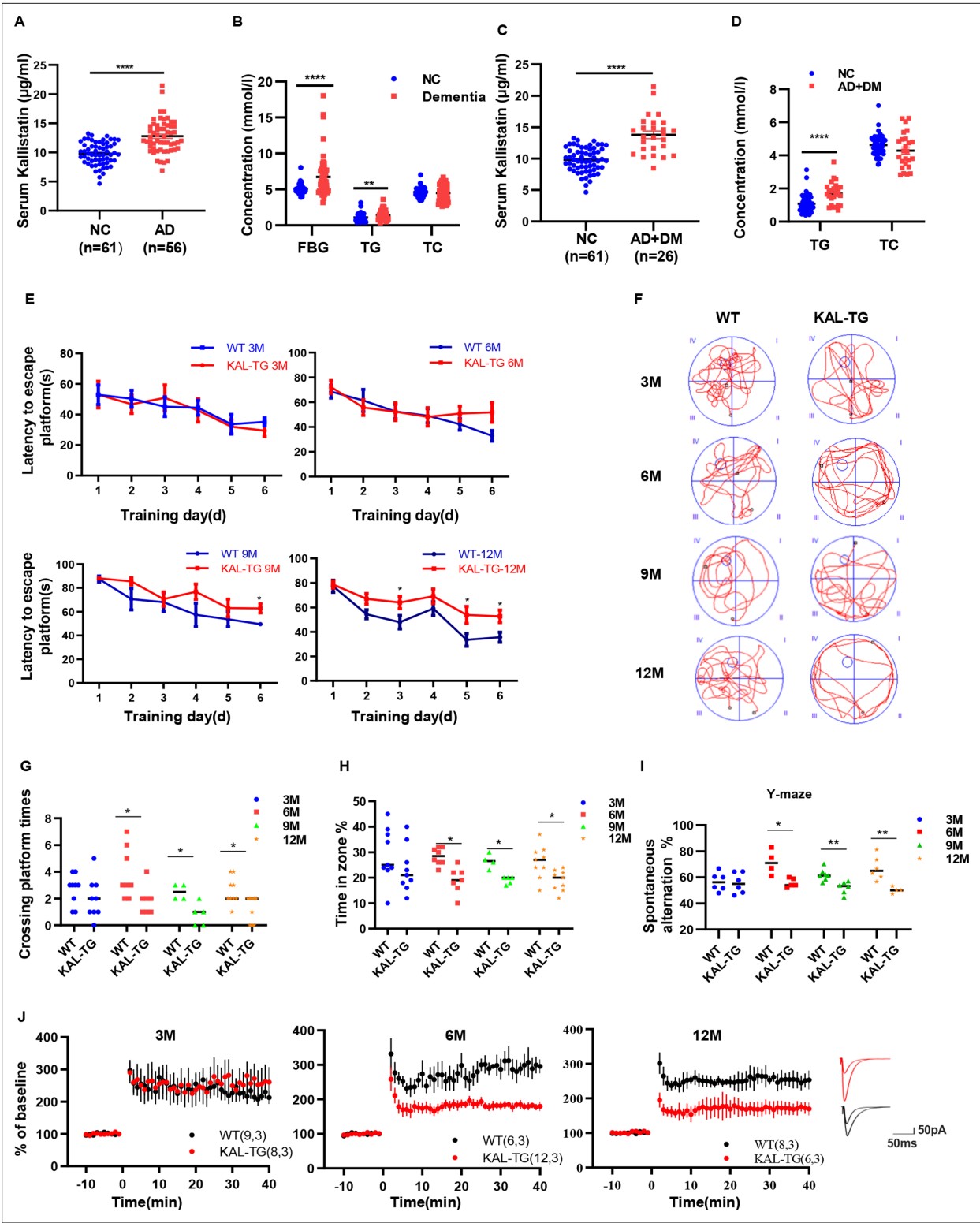

**Figure 1.** Increased Kallistatin was presented in Alzheimer's disease (AD) patients and could impair cognitive memory in mice. Serum Kallistatin (**A**), fasting blood glucose (FBG), triglyceride (TG), and total cholesterol (TC) (**B**) of AD patients and their corresponding normal control subjects. Serum Kallistatin (**C**), TG, and TC (**D**) of AD patients with DM and their corresponding normal control subjects (Student's *t*-test). (**E–J**) The behavioral performance of KAL-TG mice was assessed through the Morris water maze test, Y-maze test, and electrophysiology. (**E**) The escape latency time of different months of KAL-TG mice (3, 6, 9, and 12 M) and corresponding WT mice was presented during 1–6 days (two-way ANOVA). Cognitive functions were evaluated by spatial probe test at day 7 (Student's *t*-test), the representative each group mice traces were shown (**F**), then analyzing each group

*Figure 1 continued*

mice crossing platform times (**G**) and time percent in the targeted area (**H**), *n* = 4–9 per group. (**I**) Spontaneous alternation of Y-maze test. (**J**) Long-term potentiation (LTP) was measured by whole-cell voltage-clamp recordings of CA1 neurons in acute hippocampal slices of KAL-TG (3, 6, and 12 M) and WT mice (Student's *t*-test, *n* = 6–12 cells from 3 mice per group). Error bars represent the standard deviation (SD); *p < 0.05; **p < 0.01; ***p < 0.0001.

The online version of this article includes the following source data and figure supplement(s) for figure 1:

**Figure supplement 1.** Kallistatin is associated with Alzheimer's disease.

**Figure supplement 1—source data 1.** Kallistatin was upregulated in the hippocampal tissue of SAMP8 mice.

**Figure supplement 1—source data 2.** Kallistatin was upregulated in the hippocampal tissue of SAMP8 mice.

were much greater than those in age-matched WT mice (*Figures 2A–C and 3A–D*). Consistent with these results, ELISA detection of the Aβ42 content in hippocampal tissue revealed that Aβ production was extraordinarily increased in KAL-TG mice compared with WT mice (*Figure 2D*). These results suggested that Kallistatin promoted Aβ deposition and tau phosphorylation.

## Kallistatin positively regulates Aβ generation by promoting β-secretase rather than γ-secretase

Western blot and ELISA analyses revealed that the Aβ levels in primary hippocampal neurons (immunofluorescence identified with the neural marker MAP2, *Figure 2—figure supplement 1A*) infected with the Kallistatin adenovirus were greater than those in the control groups (*Figure 2E–G*), as were those in the HT22 cells (*Figure 2—figure supplement 1B–D*). Amyloid-β precursor protein (APP) undergoes proteolytic processing to generate peptide fragments (*De Strooper and Annaert, 2000*). β-Secretase (BACE1) and γ-secretase, which are composed of presenilin 1 (PS1), nicastrin, and Pen-2, are crucial enzymes for Aβ generation (*LaFerla and Oddo, 2005; Scheuner et al., 1996*). Compared with those in WT mice, BACE1 protein and mRNA levels were greater in the hippocampus of KAL-TG mice (*Figure 4A–C*, *Figure 4—figure supplement 1A*), whereas no significant difference in APP, PS1, or α-secretase (*Adam9, Adam10,* or *Adam17*) expression was detected (*Figure 4A*, *Figure 4—figure supplement 1B*). Consistent with the above results, the activity of BACE1 increased (*Figure 4D*), whereas PS1 activity did not change (*Figure 4—figure supplement 1C*). Similarly, the expression and activity of BACE1 were found to be increased in primary mouse neurons and HT22 cells transfected with Kallistatin adenovirus (*Figure 5A–C*, *Figure 5—figure supplement 1A–C*), while PS1 expression and activity remained unchanged (*Figure 5A, D*, *Figure 5—figure supplement 1A*). Additionally, the effect of Kallistatin was attenuated by the BACE1 inhibitor verubecestat or si*Bace1* 03, which was the most effective (*Figure 5E, F*, *Figure 5—figure supplement 1D*). These results indicate that Kallistatin can promote Aβ generation through the upregulation of BACE1 expression.

## Kallistatin suppresses PPARγ activation to promote BACE1 expression

The transcription factors SP1, YY1, and PPAR reportedly regulate BACE1 expression at the transcriptional level. Among them, PPARγ can downregulate BACE1 expression (*Christensen et al., 2004; Lin et al., 2016; Nowak et al., 2006*). PPARγ decreased in the hippocampal tissue of KAL-TG mice, as detected by western blot and immunohistochemical analysis (*Figure 5G–J*). However, no significant differences in YY1 or SP1 expression were detected (*Figure 5G*). In addition, Kallistatin downregulated the expression of PPARγ in primary hippocampal neurons and HT22 cells, thus increasing BACE1 and Aβ expression (*Figure 5K*, *Figure 5—figure supplement 1E*). Treatment with rosiglitazone, a PPARγ agonist, reversed the decrease in PPARγ caused by Kallistatin (*Figure 5K, L*). Predictably, rosiglitazone inhibited the ability of Kallistatin to promote BACE1 and Aβ (*Figure 5*, *Figure 5—figure supplement 1E*).

## Kallistatin promotes Aβ production via direct binding to the Notch1 receptor and activating the Notch1 pathway

Quantitative assessments in mice models indicated that Notch1 was highly expressed in the hippocampal tissues of KAL-TG mice (*Figure 6A–D*). Furthermore, Notch1 expression was upregulated in primary hippocampal neurons and HT22 cells infected with adenovirus expressing Kallistatin in vitro (*Figure 6—figure supplement 1A, B*). Additionally, Kallistatin could directly bind to the Notch1 receptor and activate the Notch1 pathway (*Figure 6E, F*, *Figure 6—figure supplement 1C, D*).

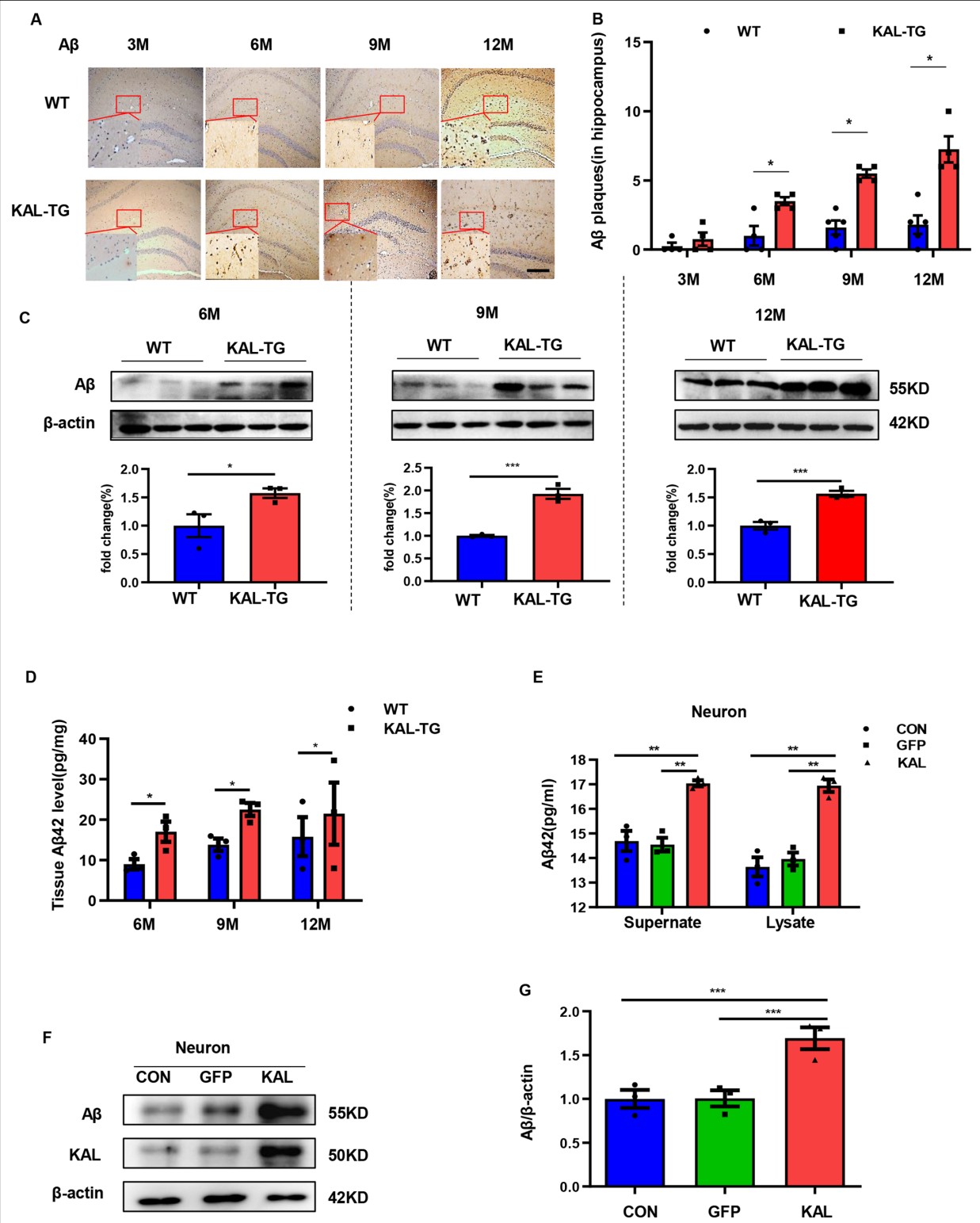

**Figure 2.** Kallistatin promoted Aβ generation. (**A, B**) Immunohistochemistry staining of Aβ (**A**) was carried out in KAL-TG and WT mice hippocampal tissue. *Scale bar*, 100 μm. The statistical analysis of Aβ plaques (**B**) in hippocampal tissue of KAL-TG and WT mice, *n* = 4–5 per group. (**C**) Protein levels of Aβ were tested by western blot analysis in hippocampal tissue, *n* = 3 per group, then statistically analyzed the above results. (**D**) Hippocampal tissue Aβ42 contents were performed by ELISA in KAL-TG and WT groups, *n* = 3 per group. (**E**) Primary mouse neurons were isolated, then infected with adenovirus to overexpress Kallistatin for 72 hr. Aβ42 concentration of primary hippocampal neurons supernate and cell lysate was quantified by ELISA, *n* = 3 per group. (**F, G**) Western blot analysis of Aβ protein level in primary hippocampal neurons infected with overexpressing Kallistatin adenovirus and

*Figure 2 continued on next page*

*Figure 2 continued*

control groups, then statistical analysis of Aβ protein levels, $n$ = 3 per group. Error bars represent the standard deviation (SD); *p < 0.05; **p < 0.01; ***p < 0.001; Student's $t$-test.

The online version of this article includes the following source data and figure supplement(s) for figure 2:

**Source data 1.** Kallistatin promoted Aβ generation.

**Source data 2.** Kallistatin promoted Aβ generation.

**Figure supplement 1.** Kallistatin promoted Aβ generation in HT22 cells.

**Figure supplement 1—source data 1.** Kallistatin promoted Aβ generation in HT22 cells.

**Figure supplement 1—source data 2.** Kallistatin promoted Aβ generation in HT22 cells.

Treatment with si*Notch1* 03, which was the most effective (*Figure 6—figure supplement 1E*), to knock down Notch1 inhibited the effect of Kallistatin on the activation of the Notch1 signaling pathway, leading to the downregulation of HES1, upregulation of PPARγ, and downregulation of BACE1 and Aβ (*Figure 6G*). HES1, an essential downstream effector of the Notch1 signaling pathway, has been reported to suppress the expression of PPARγ in neurons (*Maniati et al., 2011*; *Herzig et al., 2003*). Similarly, *Hes1* shRNA 1, the most effective (*Figure 6—figure supplement 1F*), upregulated PPARγ and decreased the production of BACE1 and Aβ when neurons were infected with adenovirus to overexpress Kallistatin (*Figure 6H*). These results suggest that Kallistatin promotes Aβ production via direct binding to the Notch1 receptor and activating the Notch1 pathway.

## Kallistatin promotes the phosphorylation of tau by activating the Wnt signaling pathway

Glycogen synthase kinase 3-β (GSK-3β) is a crucial element in the phosphorylation of tau (*Kanno et al., 2016*). When Wnt signaling is activated, the LRP6 PPPSPxS motif can directly interact with GSK-3β and phosphorylate it (*Piao et al., 2008*). Consequently, when Wnt signaling is inhibited, GSK-3β becomes activated and dephosphorylated, allowing nonphosphorylated GSK-3β to add phosphate groups to serine/threonine residues of tau (*Boonen et al., 2009*). Kallistatin has already been reported as a competitive inhibitor of the canonical Wnt signaling pathway (*Liu et al., 2013*). Consistent with previous reports, western analysis in both in vivo and in vitro models demonstrated that GSK-3β was activated in the hippocampus of KAL-TG mice (*Figure 7A, B*). Moreover, an increase in tau phosphorylation was observed with the activation of GSK-3β induced by Kallistatin overexpression (*Figure 7C, D*, *Figure 7—figure supplement 1A*), which was reversed by LiCl, an inhibitor of GSK-3β (*Figure 7E, F*, *Figure 7—figure supplement 1B*). These findings confirmed that Kallistatin promoted tau phosphorylation by activating the Wnt signaling pathway.

## Fenofibrate alleviates memory and cognitive impairment in KAL-TG mice

Hyperlipidemia and hyperlipidemia account for the development of AD (*Kubis-Kubiak et al., 2022*; *Rivas-Domínguez et al., 2023*). Here, a hypolipidemic drug (fenofibrate) and a hypoglycemic drug (rosiglitazone) were used to treat KAL-TG mice (*Figure 8A*). Compared with that of the KAL-TG group, the behavioral performance of the treated group was improved, as measured by the MWM and Y-maze tests. The latency to reach the escape platform on the fifth training day was significantly decreased (*Figure 8B*), and the number of platform crossings (*Figure 8C*), percentage of time spent (*Figure 8D*), and spontaneous alternation (*Figure 8F*) were significantly greater in the fenofibrate-treated group than in the rosiglitazone-treated group. Similarly, the path trace heatmap indicated that the mice in the fenofibrate-treated group stayed in the target quadrant longer (*Figure 8E*). In addition, decreased serum Kallistatin levels, Aβ and BACE1 levels, phosphorylation of tau, and activation of GSK3β were detected in the fenofibrate-treated KAL-TG group (*Figure 8G–K*). However, there was no significant difference between the rosiglitazone-treated group and the KAL-TG group (*Figure 8G–K*).

## Mechanism summary

In patients with metabolic abnormality-related AD, the concentration of Kallistatin is elevated, which could increase Aβ deposition through the Notch1/HES1/PPARγ/BACE1 pathway and induce tau

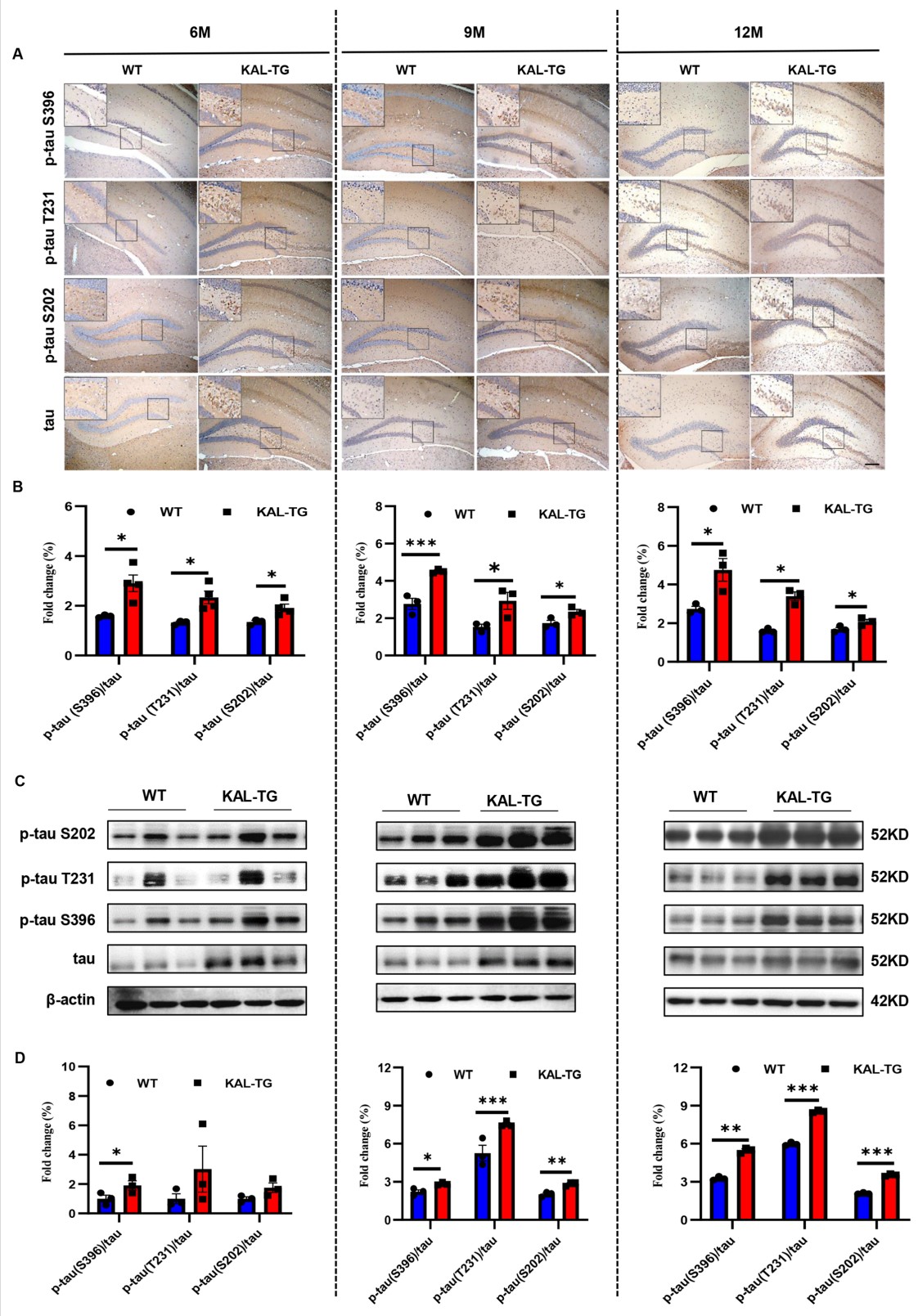

**Figure 3.** Kallistatin promoted tau phosphorylation. (**A, B**) Immunohistochemistry staining of phosphorylated tau (p-tau S396, p-tau T231, and p-tau S202) and tau (**A**) was carried out in KAL-TG and WT mice hippocampal tissue. Scale bar, 100 μm. The statistical analysis of phosphorylated tau (**B**) in hippocampal tissue of KAL-TG and WT mice, *n* = 3 per group. (**C, D**) Protein levels of phosphorylated tau (p-tau S396, p-tau T231, and p-tau S202) and

*Figure 3 continued on next page*

*Figure 3 continued*

tau were tested by western blot analysis in hippocampal tissue, then statistically analyzed the above results. Error bars represent the standard deviation (SD); *p < 0.05; **p < 0.01; ***p < 0.001; Student's *t*-test.

The online version of this article includes the following source data for figure 3:

**Source data 1.** Kallistatin promoted tau phosphorylation.

**Source data 2.** Kallistatin promoted tau phosphorylation.

hyperphosphorylation by activating GSK-3β. Consequently, elevated Kallistatin impaired cognitive memory by inducing Aβ deposition and tau hyperphosphorylation (*Figure 7—figure supplement 1C*).

## Discussion

A molecular and biochemical approach demonstrated that Kallistatin is a novel regulator of Aβ plaque accumulation, tau protein hyperphosphorylation, and metabolic abnormality-related cognitive memory impairment. It was shown that Kallistatin levels were increased in the serum of patients with AD and diabetes-related AD, as well as in the hippocampus of AD model mice. Additionally, the KAL-TG mice exhibited cognitive memory impairment and lower LTP in hippocampal CA1 neurons, along with increased Aβ deposition and tau phosphorylation. Mechanistically, Kallistatin transcriptionally upregulates *Bace1* expression by suppressing the transcriptional repressor PPARγ, leading to Aβ cleavage and production. Most importantly, the experimental findings revealed that Kallistatin could bind directly to the Notch1 receptor and activate the Notch1/HES1 pathway, causing a decrease in PPARγ, overproduction of BACE1, and increased Aβ42 generation. Moreover, Kallistatin can induce tau phosphorylation by activating GSK-3β, which results from the inhibition of LRP6. Finally, prolonged stimulation with high concentrations of Kallistatin impaired cognitive memory in mice. Finally, the hypolipidemic drug fenofibrate decreased Aβ expression, tau phosphorylation, and the serum Kallistatin level in KAL-TG mice, alleviating memory and cognitive impairment. For the first time, these observations establish an association between high Kallistatin levels and metabolic abnormality-related AD and provide a new drug candidate (fenofibrate) for AD patients with metabolic syndromes.

Increasing evidence suggests that diabetes mellitus and AD are closely linked. About 80% of AD patients are insulin resistant or have T2DM *de la Monte, 2014*; additionally, T2DM patients have a higher risk of up to 73% dementia than healthy controls do (*Koekkoek et al., 2015*). In line with these observations, the process of cognitive decline in T2DM patients appears to begin in the prediabetic phase of insulin resistance (*Biessels and Reagan, 2015*; *Xu et al., 2010*). A GAD disease enrichment analysis of differentially expressed genes in the neurons of T2DM patients and normal controls revealed a close relationship between AD and T2DM. Additionally, PFAM analysis identified an enrichment of the Serpin family protein domain (*Figure 1—figure supplement 1A, B*). Previous investigations have documented elevated circulating Kallistatin concentrations, a serine protease inhibitor (serpin) family member, in individuals with T2DM, with concomitant metabolic disturbances suggesting its potential involvement in the pathophysiological processes underlying metabolic disorders (*El-Asrar et al., 2015*; *Feng et al., 2019*). While the mechanistic association between Kallistatin and AD pathogenesis remains uncharacterized in the existing literature, emerging evidence posits Kallistatin as a pivotal molecular mediator bridging neurodegenerative pathology and metabolic dysregulation, potentially orchestrating the pathophysiological crosstalk between AD and T2DM. Kallistatin levels (*Figure 1A–D*, *Figure 1—figure supplement 1C, D*). Additionally, KAL-TG mice showed impaired memory, cognitive function, and synaptic plasticity (*Figure 1E–J*). These results suggest that Kallistatin represents a novel connection between AD and diabetes.

A hallmark of AD is the aggregation of Aβ into amyloid plaques and tau phosphorylation in patients' brains. Aβ, a small peptide with a high propensity to form aggregates, is widely believed to be central and initial to the pathogenesis of this disease (*Selkoe, 2000*). Correspondingly, it was discovered that Kallistatin could lead to Aβ overproduction through the Notch1/HES1/PPARγ/BACE1 signaling pathway. GSK-3β, a vital kinase that regulates the process of tau phosphorylation (*Kanno et al., 2016*), can be activated by inhibiting the Wnt signaling pathway. Kallistatin has been identified as a competitive inhibitor of LRP6, the Wnt receptor (*Liu et al., 2013*). Consistent with previous reports, Kallistatin increased tau phosphorylation by activating GSK-3β. It was demonstrated for the

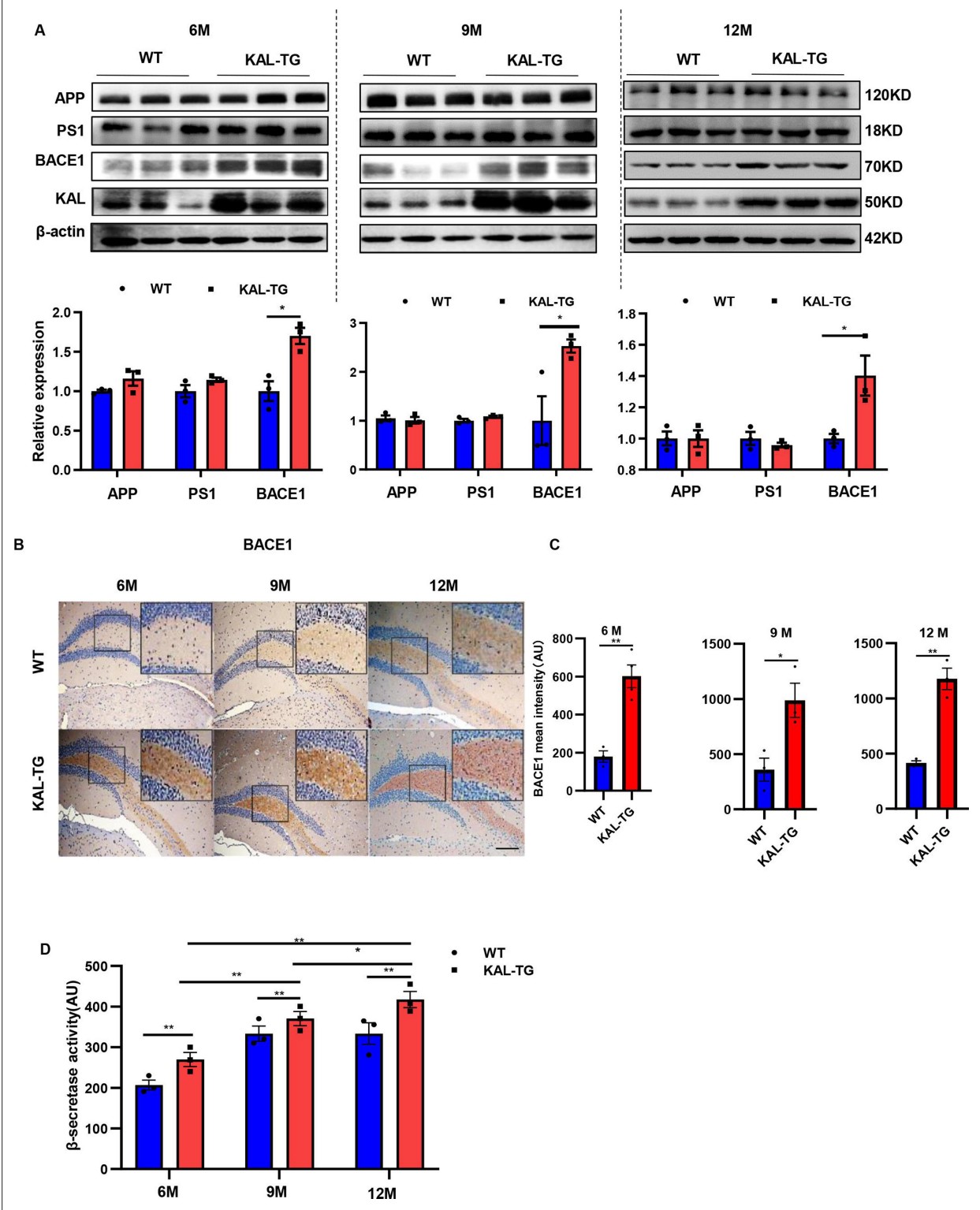

**Figure 4.** Kallistatin-transgenic mice exhibited increased BACE1 expression and activity in the hippocampus. (**A**) Western blot analysis of relevant proteins, such as APP, PS1, and BACE1 during Aβ generation in hippocampal tissue of each time point (6, 9, and 12 M) KAL-TG mice and corresponding WT control groups, *n* = 3 per group, then statistical analysis of APP, PS1, and BACE1 protein levels. (**B**) Immunohistochemistry staining of BACE1 was carried out in KAL-TG and WT mice hippocampal tissue at each time point (6, 9, and 12 M). *n* = 3–5 per group. *Scale bar*, 100 μm. (**C**) Statistical analysis of BACE1 immunohistochemistry staining, *n* = 3–4 per group. (**D**) ELISA measured the β-secretase activity of each group's hippocampal tissue, *n* = 3 per group. Error bars represent the standard deviation (SD); *p < 0.05; **p < 0.01; Student's *t*-test.

*Figure 4 continued on next page*

Figure 4 continued

The online version of this article includes the following source data and figure supplement(s) for figure 4:

**Source data 1.** Kallistatin-transgenic mice exhibited increased BACE1 expression.

**Source data 2.** Kallistatin-transgenic mice exhibited increased BACE1 expression.

**Figure supplement 1.** Kallistatin promotes β-secretase rather than α- and γ-secretase.

first time that Kallistatin promoted AD by increasing Aβ production and tau phosphorylation in the central nervous system.

Previous studies have shown that Notch signaling is closely related to AD. For example, a *Notch* mutation was reported to cause AD-like pathology (*Thijs et al., 2003*). In addition, the level of Notch1 was found to be increased in AD patients (*Brai et al., 2016*). In addition, the Notch1/HES1 signaling pathway has been reported to suppress the expression of PPARγ (*Liu et al., 2021*). The precious findings demonstrated that Kallistatin can activate Notch1 signaling (*Feng et al., 2019*). Consequently, quantitative assessments in both in vivo and in vitro models revealed mechanistically synchronized activation of Notch1 signaling by Kallistatin (*Figure 6*, *Figure 6—figure supplement 1*), as was Aβ deposition. Notch signaling is initiated by receptor–ligand interactions at the cell surface. In mammals, there are five ligands encoded by JAG1, JAG2, DLL1, DLL3, and DLL4 (*D'Souza et al., 2010*). Here, the findings revealed that Kallistatin could activate Notch1 by binding directly to it, indicating that Kallistatin is a new ligand of the Notch1 receptor.

The treatment of AD has always been a prominent and challenging issue in neurology. Multiple strategies have been proposed to reduce the pathogenicity of Aβ and tau. Unfortunately, several Aβ-targeted therapies tested in phase III clinical trials have failed to slow cognitive decline, although they can effectively reduce the Aβ load (*Salloway et al., 2014*; *Ostrowitzki et al., 2017*; *Honig et al., 2018*; *Henley et al., 2019*; *Egan et al., 2019*; *Panza et al., 2019*). BACE1 inhibitors have not only failed to improve the cognitive function of AD patients but have also resulted in clinical deterioration and liver function impairment (*Wessels et al., 2020*; *Novak et al., 2020*; *Timmers et al., 2018*). Two possible reasons for the failure of clinical trials with BACE1 inhibitors are: first, the reduction in BACE1 activity could lead to the accumulation of full-length APP *Nigam et al., 2016*; second, the size of the BACE1 active site is relatively large, and the use of a small molecule may not be sufficient to occupy the active site (*Citron, 2002*). Consequently, synaptic damage caused by BACE1 inhibitors or their insufficient effect may lead to the failure of clinical trials. Therapeutic strategies targeting tau include tau aggregation blockers (TRx0014 and TRx0237), antibody vaccine therapy (e.g., RO7105705 and BIIB092), the inhibition of tau phosphorylation (Anavex2-73), and the use of microtubule stabilizers (Anavex2-73) (*Long and Holtzman, 2019*). Some of these drugs have been partially discontinued, while others are still undergoing clinical testing and have shown protective benefits. Nonetheless, several obstacles remain to the commercialization of tau treatments when they reach maturity.

Because of the failure of clinical trials, some researchers have proposed alternative options for AD therapeutics to address modifiable risk factors for the development of AD, such as type 2 diabetes (*Livingston et al., 2017*; *Butterfield et al., 2014*; *Arnold et al., 2018*). Previous studies revealed that Kallistatin is a multifunctional protein strongly associated with diabetes and that a Kallistatin neutralizing antibody improves diabetic wound healing (*Feng et al., 2019*). The findings of the current study demonstrated that Kallistatin-induced memory-related cognitive dysfunction by promoting Aβ deposition and tau phosphorylation. Thus, it is speculated that increased Kallistatin could be a promising candidate for T2DM-related AD therapy. PPARγ is a ligand-activated transcription factor and a master modulator of glucose and lipid metabolism, organelle differentiation, and inflammation (*Zhao et al., 2016*; *Guo et al., 2017*). Growing evidence revealed that PPARγ agonists (rosiglitazone) could rescue memory impairment of AD model mice (*Toledo and Inestrosa, 2010*; *Escribano et al., 2010*; *O'Reilly and Lynch, 2012*). In clinical trials, it is controversial whether rosiglitazone has a protective effect on memory cognitive function (*Watson et al., 2005*; *Tzimopoulou et al., 2010*; *Harrington et al., 2011*). The experimental data revealed that while Aβ expression demonstrated a downward trajectory, KAL-TG mice exhibited no significant alterations in memory or cognitive function following 1-month rosiglitazone administration (*Figure 8*). These observations could potentially be attributed to insufficient therapeutic duration and persistent unmodified Kallistatin concentrations.

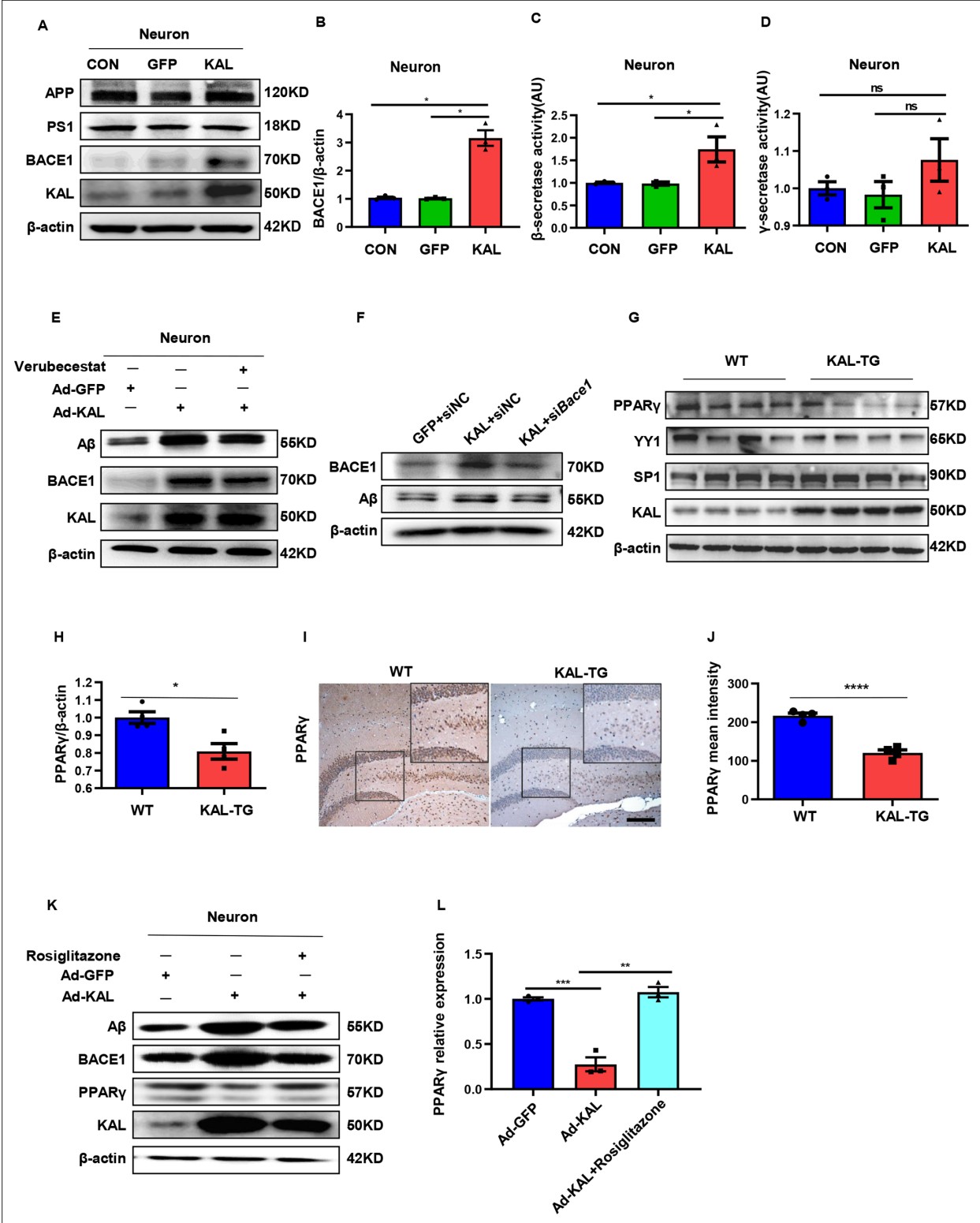

**Figure 5.** In vitro, Kallistatin promoted BACE1 expression to augment Aβ by suppressing PPARγ activation. (**A**) The relevant protein levels in primary mouse neurons infected with overexpressing Kallistatin adenovirus during Aβ generation were determined by western blot analysis. (**B**) Statistical analysis of BACE1 expression in primary neurons. β-Secretase (**C**) and γ-secretase (**D**) activity of primary hippocampal neurons infected with overexpressing Kallistatin adenovirus and control adenovirus. was measured by ELISA. (**E**) Primary hippocampal neurons were treated with BACE1 inhibitor verubecestat (50 nM), then infected with adenovirus to overexpress Kallistatin. Western blot analysis of Aβ, BACE1, and Kallistatin protein levels; β-actin served as a loading control. (**F**) HT22 cells were infected with BACE1 siRNA, then infected with adenovirus to overexpress Kallistatin.

*Figure 5 continued on next page*

*Figure 5 continued*

Western blot analysis of Aβ and BACE1 protein levels, β-actin served as a loading control. (**G**) The relevant proteins involved in BACE1 transcriptional expressions, such as PPARγ, YY1, and SP1, were measured by western blot analysis in hippocampal tissue. β-Actin served as a loading control. (**H**) Statistical analysis of PPARγ in hippocampal tissue of each group. (**I**) The representative diagrams of PPARγ expression in hippocampal tissue were presented in the above graphs. *Scale bar*, 100 μm. (**J**) Statistical analysis of PPARγ immunohistochemistry staining in hippocampal tissue of each group, $n$ = 3–4 per group. (**K**) Primary hippocampal neurons were treated with PPARγ agonist rosiglitazone (10 nM) for 12 hr, then infected with adenovirus to overexpress Kallistatin for 72 hr. Western blot analysis of Aβ and BACE1 protein levels. β-Actin served as a loading control. (**L**) Statistical analysis of PPARγ protein levels in each group. Error bars represent the standard deviation (SD); *$p < 0.05$; **$p < 0.01$; ***$p < 0.001$; ****$p < 0.0001$; ns means no significant difference; Student's *t*-test.

The online version of this article includes the following source data and figure supplement(s) for figure 5:

**Source data 1.** Original membranes corresponding to *Figure 5A, E–G, K*.

**Source data 2.** Original membranes corresponding to *Figure 5A, E–G, K*.

**Figure supplement 1.** Kallistatin increased BACE1 expression in HT22 cells.

**Figure supplement 1—source data 1.** Original membranes corresponding to *Figure 5—figure supplement 1A, D, E*.

**Figure supplement 1—source data 2.** Original membranes corresponding to *Figure 5—figure supplement 1A, D, E*.

Fenofibrate is a fibric acid derivative for clinically lowering blood lipids, mainly triglycerides (*Keating and Croom, 2007*). Studies showed that fenofibrate could prevent memory disturbances, maintain hippocampal neurogenesis, and protect against Parkinson's disease (PD) (*Barbiero et al., 2014*; *Ouk et al., 2014*). Specifically, fenofibrate has a neuroprotective effect on memory impairment induced by Aβ through targeting α- and β- secretase (*Assaf et al., 2020*). Recently, the evidence from our previous study proved that fenofibrate could repair the disrupted glutamine–glutamate cycle by upregulating glutamine synthetase, while there is currently no fenofibrate treatment of AD in clinical trials (*Long et al., 2024*). The experimental findings demonstrated that pharmacological intervention with fenofibrate exerted therapeutic effects on cognitive deficits and memory dysfunction in KAL-TG mice. In addition, Aβ, BACE1, phosphorylated tau, and serum Kallistatin levels of KAL-TG mice could be downregulated after the treatment of fenofibrate. All of these suggested that fenofibrate might be helpful for metabolic abnormalities-related AD patients. Therefore, fenofibrate administration in patients with metabolic syndrome plays an early role in preventing and treating AD.

Collective evidence from this investigation indicates elevated circulating Kallistatin levels in individuals with comorbid AD and diabetes mellitus, suggesting a potential compensatory mechanism within the neuroendocrine regulatory axis. In addition, it is the first time that Kallistatin was discovered positively regulating Aβ42 through Notch1/HES1/PPARγ/BACE1 and increased phosphorylated tau through inhibition of the Wnt signaling pathway. Kallistatin might play a crucial role in linking diabetes and cognitive memory deterioration. Moreover, fenofibrate could decrease the serum Kallistatin level, BACE1, Aβ, and phosphorylated tau of KAL-TG mice, leading to alleviating memory and cognitive impairment. These findings might provide new insight into AD and possibly other neurodegenerative disorders.

## Materials and methods

### Human subjects

The study was approved by the experimental ethics committee of Guangdong Academy of Medical Sciences and Sun Yat-sen University and carried out in strict accordance with the ethical principles, and each participant was provided written informed consent before collecting samples. The study was performed in accordance with the 1964 Declaration of Helsinki and later amendments. Sixty-one normal human samples, 56 AD patient samples, of whom 36 normal human samples were from the Zhongshan City People's Hospital; 14 normal human samples and 18 AD patient samples were from Zhongshan Third People's Hospital; 11 normal human samples and 14 AD patient samples were from Sun Yat-sen Memorial Hospital, 24 AD patient samples were from Guangdong Provincial People's Hospital. AD patients were clinically diagnosed according to ICD-10 (International Classification of Diseases) and NINCDS-ADRDA (the National Institute of Neurological and Communicative Disorders and Stroke and the Alzheimer's Disease and Related Disorders Association) criteria, and 20 AD patient samples were clinically diagnosed according to MMSE (Mini-Mental State Examination) and

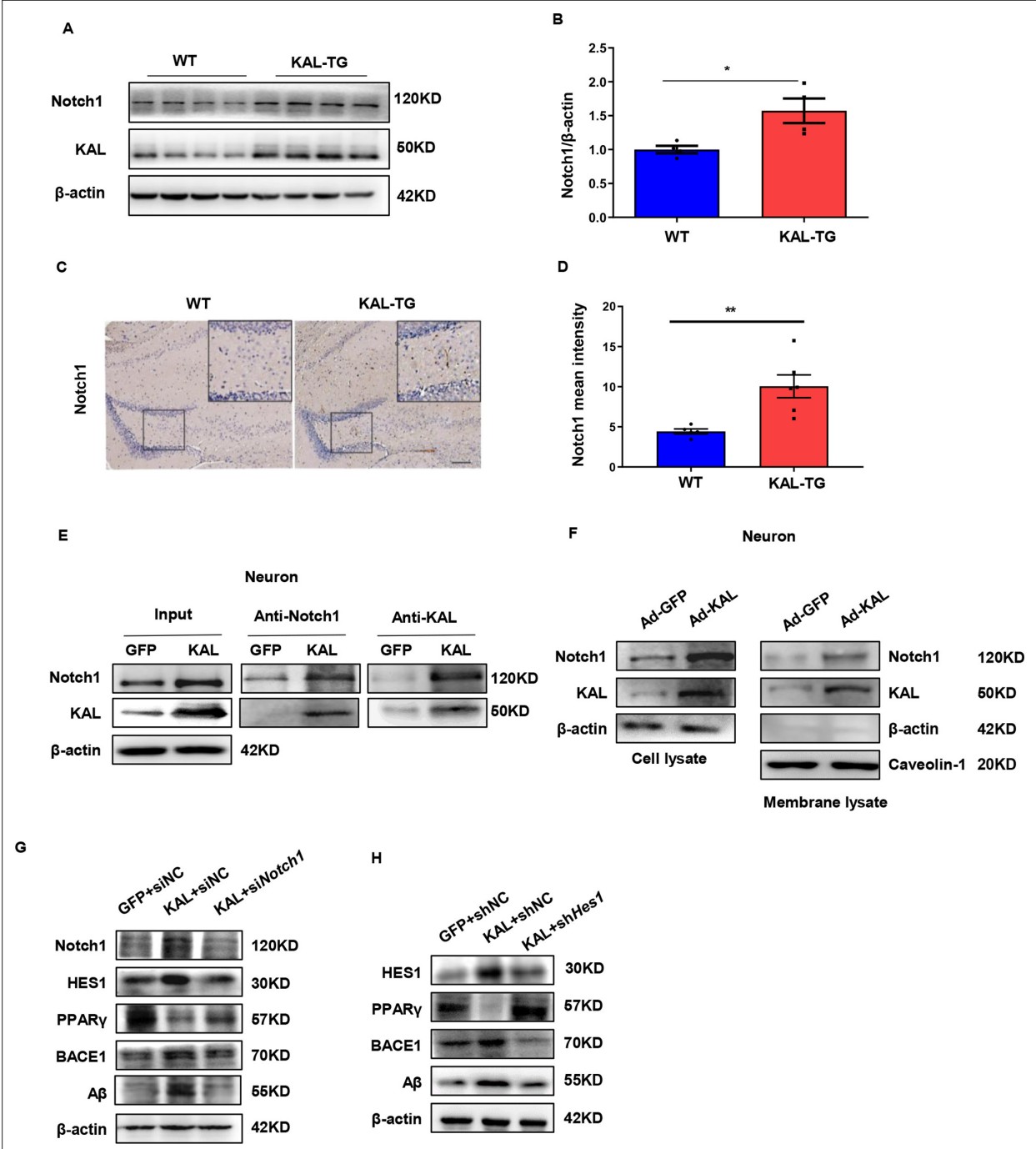

**Figure 6.** Kallistatin directly bonded to the Notch1 receptor, which activated the Notch1 pathway to promote Aβ production. (**A**) Notch1 expression was measured by western blot analysis in hippocampal tissue. β-Actin served as a loading control. (**B**) Statistical analysis of Notch1 in hippocampal tissue of each group. (**C**) The representative diagrams of Nocth1 expression in hippocampal tissue were presented in the above graphs. Scale bar, 100 μm. (**D**) Statistical analysis of Notch1 immunohistochemistry staining in hippocampal tissue of each group. Primary hippocampal neurons were infected with overexpressing Kallistatin adenovirus for 72 hr, then Co-IP analysis (**E**) and membrane extraction experiment (**F**) was performed to verify whether Kallistatin can bind to the Notch1 receptor. β-Actin served as a loading control. (**G, H**) HT22 cells were treated with siRNA (Notch1) and shRNA (HES1) to knock down Notch1 and HES1 for 12 hr, then infected with adenovirus to overexpress Kallistatin for 24 hr. Western blot analysis was used to detect the Notch1 signaling pathway. Error bars represent the standard deviation (SD); *p < 0.05, **p < 0.01; Student's *t*-test.

The online version of this article includes the following source data and figure supplement(s) for figure 6:

**Source data 1.** Original membranes corresponding to *Figure 6A, E–H.*

*Figure 6 continued on next page*

*Figure 6 continued*

**Source data 2.** Original membranes corresponding to *Figure 6A, E–H*.

**Figure supplement 1.** Validation of cell transfection efficiency.

**Figure supplement 1—source data 1.** Original membranes corresponding to *Figure 6—figure supplement 1A–F*.

**Figure supplement 1—source data 2.** Original membranes corresponding to *Figure 6—figure supplement 1A–F*.

were collected from Guangdong Provincial People's Hospital. All subjects' clinical characteristics were presented in *Supplementary file 1* (Tables S1 and S2).

## Experimental animals and protocols

All animal experiment procedures were carried out in an environment without specific pathogens (specific pathogen free) with the approval of the Animal Care and Use Committee of Sun Yat-sen University (approval ID: SYXK 2015-0107). The wild-type mice (WT, C57BL/6) were purchased from the Animal Center of Guangdong Province (Production License No. SCXK 2013-0002, Guangzhou, China). The SAMR1 and SAMP8 mice (7 months old) were purchased from Tianjin University of Traditional Chinese Medicine (Tianjin, China). Kallistatin-transgenic mice (KAL-TG) were C57BL/6 strain provided by Dr. Jianxing Ma (University of Oklahoma Health Sciences Center) (*McBride et al., 2014*). The KAL-TG mice genotype was identified by PCR technology (forward primer: 5′-AGGGAAGATTGT GGATTTGG-3′, reverse primer: 5′-ATGAAGATACCAGTGATGCTC-3′). KAL-TG mice aged 6 months were randomly divided into three groups: control group (KAL-TG), fenofibrate-treated group (KAL-TG-Feno, 0.3 g/kg/day), and rosiglitazone-treated group (KAL-TG-RSG, 0.005 g/kg/day). Fenofibrate (Sigma-Aldrich, Cat. No. F6020) and rosiglitazone (Selleck, Cat. No. S2556) were administered to mice by oral gavage. In three groups, the serum Kallistatin was examined in the 0 and 4 weeks after drug treatment from the blood taken from mouse orbit. In addition, the MWM and Y-maze test were performed 1 week after the second blood collection.

## Morris water maze

The KAL-TG and WT mice were employed for the MWM test, including the behavioral test, latency experiment (for 6 days), and the probe test (the 7th day). In addition, the MWM was performed as described previously (*Huang et al., 2018*). Mice were brought into the testing room and handled for 1 day before the training experiment. In the 6-day training experiment, each mouse was trained with four daily trials. The mice facing the wall were placed into the maze, exploring the maze from different directions (east, south, west, and north). This trial was completed as soon as the mouse found the platform, or 90 s elapsed. If the mice could discover and climb the submerged platform within 90 s, the system would automatically record the latency time and path immediately, and then the mouse was guided to and placed on the submerged platform for an extra 20 s. On day 7, the platform was removed, and a probe test was performed to examine the strength and integrity of the animal spatial memory 24 hr after the last testing trial. During the probe test, the mice were gently brought into the water from the fixed monitoring point, and the mice were allowed to swim for 90 s without the platform. Finally, all of the measured behavioral parameters were analyzed using SMART software.

## Y-maze test

A Y-maze test was performed to assess the mice's spatial memory. The Y maze was separated by 120°, consisting of three identical arms (30 cm long, 7 cm wide, and 15 cm high) made of blue PVC. The mice were placed first in one of the arms, and over the next 10 min, the sequence and number of their entry into the three arms were monitored. An alternation is defined when a mouse visits three straight arms (namely, ABC, BCA, or CAB, but not ABA, BAB, or CAC). Spontaneous alternation (%) = [(number of alternations)/(total number of arms − 2)] × 100.

## Electrophysiology

Hippocampal slices (300–400 μm) from KAL-TG and WT mice were cut as described (*Guo et al., 2021*). Coronal slices from hippocampus (400 μm thick) were prepared from different age groups of KAL-TG mice and their WT littermates using a tissue slicer (Vibratome 3000; Vibratome) in ice-cold dissection buffer containing the following (in mM): 212.7 sucrose, 3 KCl, 1.25 NaH$_2$PO$_4$, 3 MgCl$_2$, 1

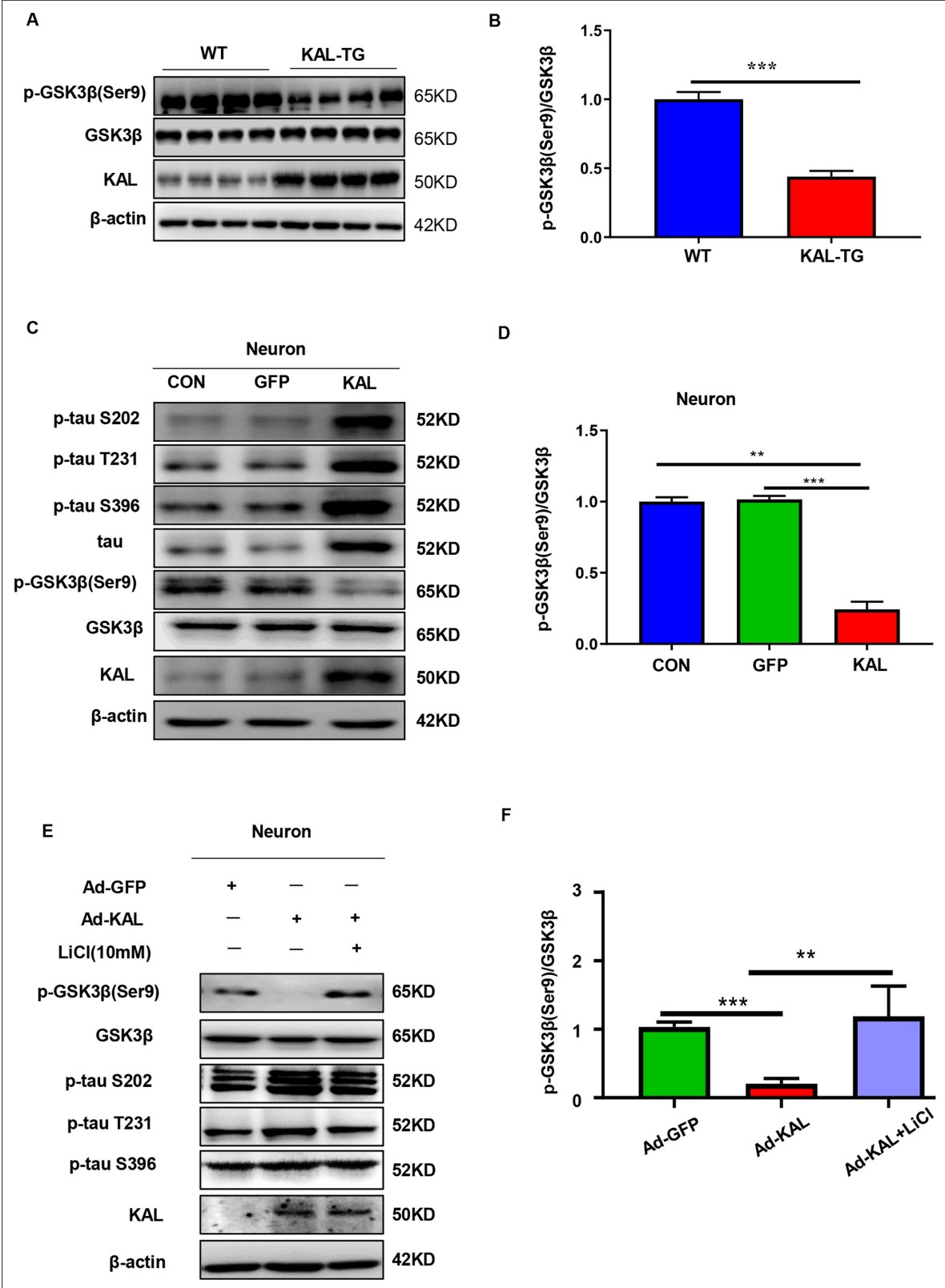

**Figure 7.** Kallistatin promoted phosphorylation of tau by suppressing the Wnt signaling pathway. (**A, B**) GSK-3β and p-GSK-3β expression was measured by western blot analysis in hippocampal tissue, then statistically analyzed the above results. (**C, D**) Primary hippocampal neurons were treated with overexpressing Kallistatin adenovirus for 72 hr, then western blot analysis was used to detect the content of GSK-3β, p-GSK-3β, tau, and p-tau (Ser9, T231, and S396), and statistically analyzed the above results. (**E, F**) Primary hippocampal neurons were treated with overexpressing Kallistatin adenovirus

*Figure 7 continued on next page*

*Figure 7 continued*

for 48 hr, then treated with LiCl (10 mM) for 24 hr. Western blot analysis was used to detect the content of GSK-3β, p-GSK-3β, and p-tau (Ser9, T231, and S396), and statistical analysis of the above results. Error bars represent the standard deviation (SD), **p < 0.01; ***p < 0.001; Student's *t*-test.

The online version of this article includes the following source data and figure supplement(s) for figure 7:

**Source data 1.** Original membranes corresponding to *Figure 7A, C, E*.

**Source data 2.** Original membranes corresponding to *Figure 7A, C, E*.

**Figure supplement 1.** Kallistatin promotes the phosphorylation of tau by activating the Wnt signaling pathway.

**Figure supplement 1—source data 1.** Kallistatin promotes the phosphorylation of tau by activating the Wnt signaling pathway.

**Figure supplement 1—source data 2.** Kallistatin promotes the phosphorylation of tau by activating the Wnt signaling pathway.

CaCl$_2$, 26 NaHCO$_3$, and 10 dextrose, bubbled with 95% O$_2$/5% CO$_2$. The slices were immediately transferred to Artificial cerebrospinal fluid (ACSF) at 35°C for 30 min before recordings. The recipe of ACSF was similar to the dissection buffer, except that sucrose was replaced with 124 mM NaCl, and the concentrations of MgCl$_2$ and CaCl$_2$ were changed to 1 and 2 mM, respectively. All recordings were performed at 28–30°C. Pyramidal cells in CA1 areas were identified visually under infrared differential interference contrast optics based on their pyramidal somata and prominent apical dendrites. Whole cell was performed using an Integrated Patch-Clamp Amplifier (Sutter Instrument, Novato, CA, USA) controlled by Igor 7 software (WaveMetrics, Portland, OR, USA) filtered at 5 kHz and sampled at 20 kHz. Igor 7 software was also used for acquisition and analysis. Only cells with series resistance <20 MΩ and input resistance >100 MΩ were studied. Cells were excluded if input resistance changed >15% or series resistance changed >10% over the experiment. A concentric bipolar stimulating electrode with a tip diameter of 125 µm (FHC MicroTargeting Electrodes) was placed in the stratum radiatum. The recording and stimulating electrode distances were kept at 50–100 µm. Patch pipettes (2–4 MΩ) were filled with the internal solution consisting of the following (in mM): 120 Cs-methylsulfonate, 10 Na-phosphocreatine, 10 HEPES, 4 ATP, 5 lidocaine *N*-ethyl bromide (QX-314), 0.5 GTP; the pH of the solution was 7.2–7.3, and the osmolarity was 270–285 mOsm.

To induce LTP, a pairing protocol was applied. In brief, conditioning stimulation consisted of 360 pulses at 2 Hz paired with continuous postsynaptic depolarization (180 s) to 0 mV. 50 µM picrotoxin was added to the recording bath to suppress excessive polysynaptic activity, and the concentration of Ca$^{2+}$ and Mg$^{2+}$ was elevated to 4 mM to reduce the recruitment of polysynaptic responses. A test pulse was delivered at 0.067 Hz to monitor baseline amplitude for 10 min before and 30 min following paired stimulation. To calculate LTP, the EPSC (Excitatory Post Synaptic Current) amplitude was normalized to the mean baseline amplitude during 10 min baseline. Potentiation was defined as the mean normalized EPSC amplitude 25–40 min after paired stimulation.

## ELISA

To quantify serum Kallistatin, the collected samples were centrifuged at 4°C for 10 min at 5000 rpm. It was detected using the KBP ELISA kit (#DY1669, R&D systems, MN, USA) as per the instructions of the manufacturer. The levels of Aβ42 in brain tissue produced from mouse primary neuron cells and HT22 cells were measured with a mouse Aβ42 Elisa Kit (27721, IBL, Germany). To measure Aβ42 in brain tissue, 0.05 g of mouse brain tissues were weighed and homogenized using 2ml PBS with a protease inhibitor (cocktail, IKM1020, Solarbio). After centrifugalization at 4°C for 30 min at 12,000 × *g*, the extracts' supernatants were analyzed using the ELISA method after total protein quantification. To quantify levels of Aβ42 produced from primary neuron cells, the cell supernatants were ultrafiltrated with an ultrafiltration tube (4-kD Millipore), centrifugalization, and testing. Cell homogenate was prepared in 1ml PBS with cocktail and quantified using the BCA method before being measured by ELISA.

## Immunohistochemistry

Tissue slices were prepared as described before (*Li et al., 2019*). The sections were incubated with Aβ (ab201060, Abcam, Cambridge, UK), BACE1 (#5606S, Cell Signaling Technology, Boston, USA), PPARγ (#2435, Cell Signaling Technology, Boston, USA), Notch1 (#3608, Cell Signaling Technology, Boston, USA), p-tau S202 (ab108387, Abcam, Cambridge, UK), p-tau T231(ab151559, Abcam, Cambridge, UK), p-tau S396 (ab109390, Abcam, Cambridge, UK), and tau (ab75714, Abcam, Cambridge, UK)

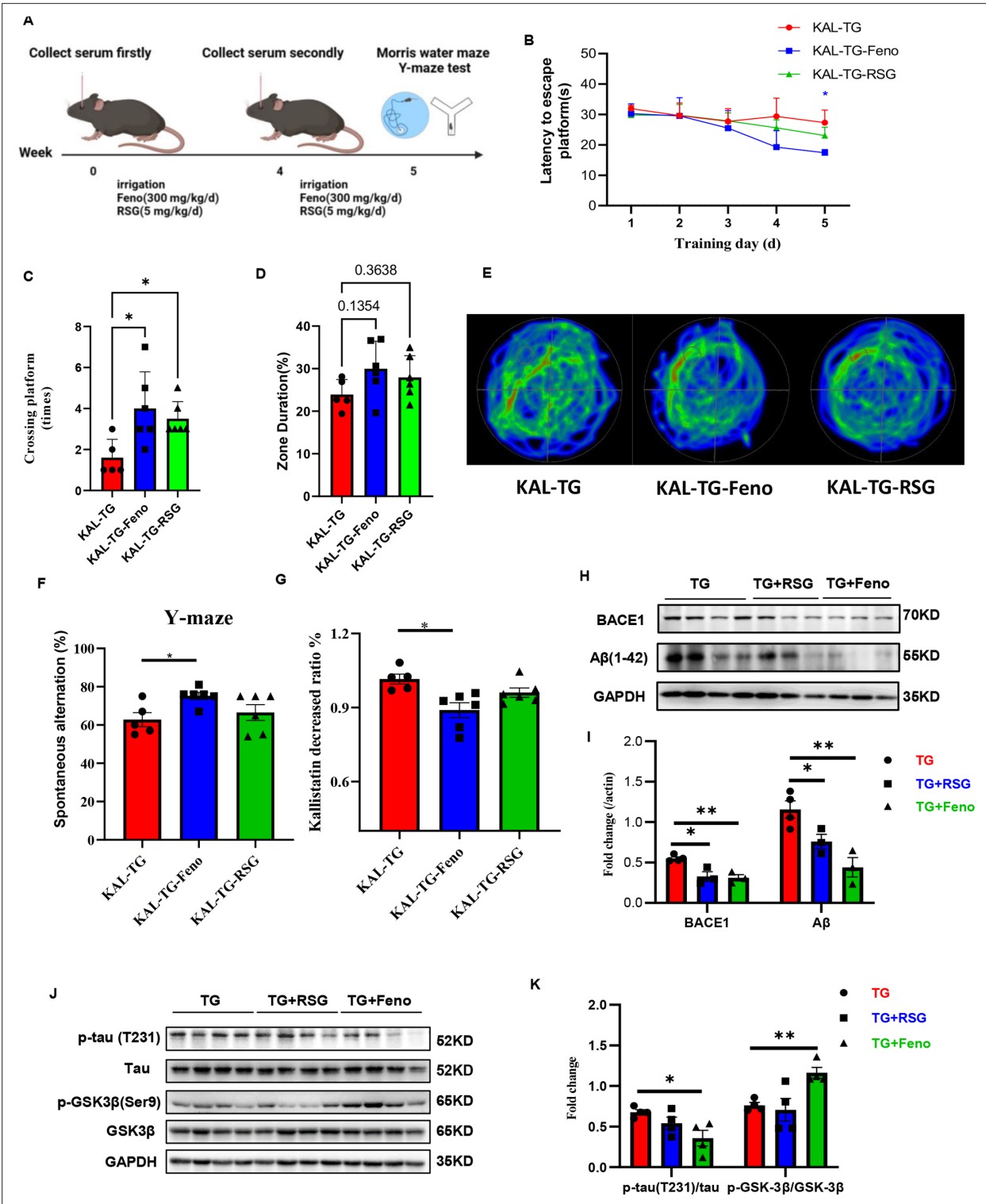

**Figure 8.** Fenofibrate could alleviate memory and cognitive impairment of KAL-TG mice. (**A**) Illustration of experimental protocols. Fenofibrate (0.3 g/kg/day × 5 weeks, i.g.) or rosiglitazone (5 mg/kg/day × 5 weeks, i.g.) was given to KAL-TG mice. The serum for Kallistatin measuring was collected at weeks 0 and 4. And the Morris water maze and Y-maze test were performed at week 5. (**B–E**) Behavioral performance was assessed through the Morris water maze test and the Y-maze test. (**B**) The escape latency time was presented during 1–5 days. (**C–E**) Cognitive functions were evaluated by spatial probe test at day 6, then analyzing each group of mice crossing platform times (**C**), time percent in the targeted area (**D**), and the path traces heatmap (**E**), n = 5–6 per group. (**F**) Spontaneous alternation of Y-maze test. (**G**) Kallistatin decreased ratio was calculated by dividing the serum Kallistatin concentration of KAL-TG mice before fenofibrate/rosiglitazone treatment by the serum Kallistatin concentration of KAL-TG mice after a month of

*Figure 8 continued*

treatment, and serum Kallistatin concentration was measured by ELISA. (**H–I**) Protein levels of Aβ and BACE1 were tested by western blot analysis in hippocampal tissue, then statistically analyzing the above results. (**J–K**) Protein levels of p-tau (231), tau, p-GSK-3β (Ser9), and GSK-3β were tested by western blot analysis in hippocampal tissue, then statistically analyzing the above results. Error bars represent the standard deviation (SD); *p < 0.05, **p < 0.01; Student's *t*-test.

The online version of this article includes the following source data for figure 8:

**Source data 1.** Fenofibrate reduced hippocampal BACE1 and Aβ levels in KAL-TG mice.

**Source data 2.** Fenofibrate activated the Wnt signaling pathway and inhibited Tau phosphorylation in the hippocampus of KAL-TG mice.

antibodies overnight at 4°C and then incubated with Alexa Fluor 488-donkey anti-rabbit IgG (H + L) (1:200, Life Technologies, Gaithersburg, MD, USA, #A21208) for 1 hr, then incubated with a biotin-conjugated secondary antibody for 30 min, followed by incubation with DAB for 10 s and hematoxylin staining for 30 s. The IHC signals were analyzed using ImageJ.

## Cell culture experiments

HT22 cells were purchased from the Cell Bank of the Chinese Academy of Sciences (Shanghai, China). HT22 cells were cultured and grown to confluence in DMEM supplemented with 10% FBS (Gibco BRL), 100 U/ml penicillin, and 100 U/ml streptomycins (Gibco BRL).

## Primary culture of hippocampal neurons

Primary neurons were obtained from the hippocampus of C57/BL6J mice at age 1–3 days. Before culturing, the newborn pup was euthanized and dipped into 70% ethanol for 3 min. First, the infant pup hippocampus was isolated with eye tweezers observed under the stereomicroscope, and excess soft tissue was removed. Second, hippocampal tissue in PBS buffer was cut up with scissors gently and blown with a 1-ml pipette until it was not visible. Next, the cell suspension was transferred to a 15-ml centrifuge tube and centrifuged at 1000 rpm for 5 min at room temperature. Cell precipitation was suspended and cultured with 2–3 ml primary neural stem cell suspension (Thermo Fisher Scientific, 21103049) in a 37°C, 5% $CO_2$ cell incubator for 3 days, changing half medium every 2 days. After 7 days, the cell suspension was transferred to a 15-ml centrifuge tube, centrifuged, and recultured with neurobasal, 10% FBS, 1:50 B27 (Thermo Fisher Scientific, A3582801), and 1:100 bFGF (Thermo Fisher Scientific, #RP-8626). One day later, the medium was changed to neurobasal (2% FBS, 1:50 B27, and 1:100 bFGF), culturing for 21 more days. The immunofluorescence technique was used with the neuron-specific marker (MAP2, #4542, Cell Signaling Technology, Boston, USA) to determine the purity of neurons.

## siRNA, shRNA, and adenovirus transfection

Notch1 siRNA and control siRNA were purchased from RiboBio (Guangzhou, China). Hes1 shRNA and control shRNA were purchased from Qingke (Guangzhou, China). Green fluorescent protein-adenovirus (Ad-GFP) and Kallistatin-adenovirus (Ad-KAL) were provided by Dr. Jianxing Ma (University of Oklahoma Health Sciences Center). According to the manufacturer's instructions, the transfections were performed at approximately 60% confluency using Lipofectamine3000 transfection reagent (Invitrogen) or RNAiMAX. After 24 hr, interference confirmation was conducted using real-time quantitative PCR (RT-qPCR) and western blot.

## RNA isolation and quantitative RT-PCR

Total RNA extraction, reverse transcription of cDNA, and RT-qPCR were performed as described previously (*Jiang et al., 2019*). *Bace1* forward: GGAGCCCTTCTTTGACTCCC; *Bace1* reverse: CAAT GATCATGCTCCCTCCCA; *Adam9* forward: GGAAGGCTCCCTACTCTCTGA; *Adam9* reverse: CAAT TCCAAAACTGGCATTCTCC; *Adam10* forward: ATGGTGTTGCCGACAGTGTTA; *Adam10* reverse: GTTTGGCACGCTGGTGTTTTT; *Adam17* forward: GGAT-CTACAGTCTGCGACACA; *Adam17* reverse: TGAAAAGCGTTCGGTACTTGAT; β-actin forward: GCACTCTTCCAGCTTCCTT; β-actin reverse: GTTG GCGTACAGGTCTTTGC.

## Western blot

Western blot was performed as described previously (*Huang et al., 2018*; *Jiang et al., 2019*). Equal amounts of protein were subjected to western blot analysis. Blots were probed with antibodies against Kallistatin (ab187656, Abcam, Cambridge, UK), Aβ (ab201060, Abcam, Cambridge, UK), Presenilin-1 (ab76083, Abcam, Cambridge, UK), BACE1 (#5606S, Cell Signaling Technology, Boston, USA), APP (#2452S, Cell Signaling Technology, Boston, USA), MAP2 (#4542, Cell Signaling Technology, Boston, USA), PPARγ (#2435, Cell Signaling Technology, Boston, USA), SP1 (#9389, Cell Signaling Technology, Boston, USA), YY1 (#46395, Cell Signaling Technology, Boston, USA), Notch1 (#3608, Cell Signaling Technology, Boston, USA), Hes1 (#11988, Cell Signaling Technology, Boston, USA), p-tau S202 (ab108387, Abcam, Cambridge, UK), p-tau T231 (ab151559, Abcam, Cambridge, UK), p-tau S396 (ab109390, Abcam, Cambridge, UK), tau (ab75714, Abcam, Cambridge, UK), GSK3β (#70109S, Cell Signaling Technology, Boston, USA), p-GSK3β Ser9 (#9323, Cell Signaling Technology, Boston, USA), β-actin (A5441-2ml, Sigma, CA, USA), Caveolin-1 (SZ02-01, Huabio, China), GAPDH (200306-7E4, Zen-bio, China), anti-Mouse (#PI200, Vector Laboratories, Burlingame, CA, USA), and anti-Rabbit (#PI1000, Vector Laboratories, Burlingame, CA, USA). The signal intensity was quantified using ImageJ (NIH).

## Statistical analysis

The results are expressed as mean ± SD. Student's *t*-test was applied for comparisons of parametric data between two groups, and one-way ANOVA followed by LSD *t*-test was used to compare differences between more than two different groups (GraphPad Prism software). A p value less than 0.05 was considered statistical significance.

## Acknowledgements

The authors acknowledge the generous provision of human serum specimens by collaborating clinical institutions including Zhongshan City People's Hospital, Zhongshan Third People's Hospital, Guangdong Provincial People's Hospital, and Sun Yat-sen Memorial Hospital for biomarker quantification and metabolic profiling. Gratitude is extended to Professor Boxing Li for expert guidance that enhanced the methodological rigor of the research.

## Additional information

### Funding

| Funder | Grant reference number | Author |
| --- | --- | --- |
| National Natural Science Foundation of China | 82273116 | Xia Yang |
| Natural Science Foundation of Guangdong Province | 2022A1515012423 | Weiwei Qi |
| National Key Research and Development Program of China | 2018YFA0800403 | Guoquan Gao |
| Guangzhou Key Laboratory for Metabolic Diseases | 20210210004 | Guoquan Gao |
| Natural Science Foundation of Guangdong Province | 2024A1515010149 | Xia Yang |

The funders had no role in study design, data collection, and interpretation, or the decision to submit the work for publication.

## Author contributions
Weiwei Qi, Yanlan Long, Data curation, Formal analysis, Writing – original draft, Project administration, Writing – review and editing; Ziming Li, Data curation, Software, Formal analysis, Methodology; Zhen Zhao, Data curation, Software, Project administration; Jinhui Shi, Software, Validation, Investigation, Methodology; Wanting Xie, Supervision, Investigation, Methodology; Laijian Wang, Resources, Software, Project administration; Yandan Tan, Visualization, Methodology; Ti Zhou, Validation, Investigation, Methodology; Minting Liang, Software, Formal analysis, Supervision, Investigation; Ping Jiang, Conceptualization, Validation, Investigation; Bin Jiang, Conceptualization, Resources, Validation; Xia Yang, Conceptualization, Funding acquisition, Validation; Guoquan Gao, Conceptualization, Funding acquisition, Validation, Writing – review and editing

## Author ORCIDs
Yanlan Long (ID) https://orcid.org/0000-0001-5164-4736
Ti Zhou (ID) https://orcid.org/0000-0001-8517-5405
Xia Yang (ID) https://orcid.org/0000-0003-1668-8121
Guoquan Gao (ID) https://orcid.org/0000-0001-8996-1470

## Ethics
All patients involved in this study gave their informed consent. The institutional review board approval was obtained from Medical Ethics of Zhongshan Medical College No. 072 in 2021.
All procedures were performed under specific pathogen free (SPF) conditions with approval from the Institutional Animal Care and Use Committee (lACUC), Sun Yat-Sen University (Approval No. SYSU IACUC 2019 B051).

Reviewer #1 (Public review): https://doi.org/10.7554/eLife.99462.3.sa1
Reviewer #2 (Public review): https://doi.org/10.7554/eLife.99462.3.sa2
Reviewer #3 (Public review): https://doi.org/10.7554/eLife.99462.3.sa3
Author response https://doi.org/10.7554/eLife.99462.3.sa4

---

# Additional files

## Supplementary files
Supplementary file 1. Clinical characteristics of Alzheimer's disease (AD) patients and AD patients with DM. *N* is an abbreviation for number; GLU is an abbreviation for glucose; TC is an abbreviation for total cholesterol; TG is an abbreviation for triglyceride; NA indicates not available; NS indicates no significance. **$p < 0.01$, ****$p < 0.0001$.

MDAR checklist

## Data availability
All data generated or analyzed during this study are included in the manuscript, supporting files and source data.

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
