## [Editor Report · eLife Assessment]

This **important** study identified a molecular mechanism linking diabetes to AD risk and the data presented are **convincing**. The authors investigated the role of kallistatin in metabolic abnormalities associated with AD and identified that Kallistatin is a key player that mediates Aβ accumulation and tau hyperphosphorylation in AD. This manuscript provides novel insights into the pathogenesis of AD, indicating that the hypolipidemic drug fenofibrate attenuates AD-like pathology in Kallistatin transgenic mice.

---

## [Referee Report · Reviewer #1 (Public review)]

Summary:

Qi and colleagues investigated the role of Kallistatin pathway in increasing hippocampal amyloid-β plaques accumulation and tau hyperpholphorylation in Alzheimer's disease, linking the increased Kallistatin level in diabetic patients with a higher risk of Alzheimer's disease development. A Kallistatin overexpressing animal model was utilized, and memory impairment was assessed using Morris water maze and Y-maze. Kallistatin-related pathway protein levels were measured in the hippocampus, and phenotypes were rescued using fenofibrate and rosiglitazone. The current study provides evidence of a novel molecular mechanism linking diabetes and Alzheimer's disease, and suggests the potential use of fenofibrate to alleviate memory impairment. However, several issues need to be addressed before further consideration.

Strengths:

The finding of this study is novel. The finding will have great impacts on diabetes and AD research. The studies were well conducted, and results convincing.

Weaknesses:

(1) The mechanism by which fenofibrate rescues memory loss in Kallistatin-transgenic mice is unclear. As a PPARα agonist, does fenofibrate target the Kallistatin pathway directly or indirectly? Please provide discussion based on literature supporting either possibility.

(2) The current study exclusively investigated hippocampus. What about other cognitive memory-related regions, such as prefrontal cortex? Including data from these regions or discussing the possibility of their involvement could provide a more comprehensive understanding of the role of Kallistatin in memory impairment.

(3) Fenofibrate rescued phenotypes in Kallistatin-transgenic mice while rosiglitazone, a PPARα agonist, did not. This result contradicts the manuscript's emphasis on a PPARα-associated mechanism. Please address this inconsistency.

(4) Most of the immunohistochemistry images are unclear. Inserts have similar magnification to the original representative images, making judgments difficult. Please provide larger inserts with higher resolution.

(5) The immunohistochemistry images in different figures were taken from different hippocampal subregions with different magnifications. Please maintain consistency, or explain why CA1, CA3 or DG was analyzed in each experiment.

(6) Figure 5B is missing a title. Please add a title to maintain consistency with other graphs.

(7) Please list statistical methods used in the figure legends, such as t-test or One way ANOVA with post-hoc tests.

Comments on revisions:

The authors have addressed the issues raised from the review. The manuscript has been revised accordingly.

---

## [Referee Report · Reviewer #2 (Public review)]

Summary:

The study links Alzheimer's disease (AD) with metabolic disorders through elevated Kallistatin levels in AD patients. Kallistatin-overexpressing mice show cognitive decline, increased Aβ and tau pathology, and impaired hippocampal function. Mechanistically, Kallistatin enhances Aβ production via Notch1 and promotes tau phosphorylation through GSK-3β activation. Fenofibrate improves cognitive deficits by reducing Aβ and tau phosphorylation in these mice, suggesting therapeutic potential in AD linked to metabolic syndromes.

Strengths:

This study presents a novel insights into Alzheimer's disease (AD) pathogenesis and provides strong evidences about the mechanistic roles of Kallistatin and the therapeutic potential of fenofibrate in AD.

It was suggested that Kallistatin is primarily produced by the liver. The study demonstrates increased Kallistatin levels in the hippocampus tissue of AD mice. They also found that Kallistatin is also increased in the liver of AD mice.

They also showed that Kallistatin directly binds to Notch1 and contributes to the activation of the Noch1-HES1 signaling pathway

---

## [Referee Report · Reviewer #3 (Public review)]

Summary:

The authors investigated the role of kallistatin in metabolic abnormalities associated with AD. They found that Kallistatin promotes Aβ production by binding to the Notch1 receptor and upregulating BACE1 expression. They identified that Kallistatin is a key player that mediates Aβ accumulation and tau hyperphosphorylation in AD.

Strengths:

This manuscript not only provides novel insights into the pathogenesis of AD, but also indicates that the hypolipidemic drug fenofibrate attenuates AD-like pathology in Kallistatin transgenic mice.

Weaknesses:

The authors did not illustrate whether the protective effect of fenofibrate against AD depends on kallistatin.

The conclusions are supported by the results.

---

## [Author Response]

The following is the authors’ response to the original reviews

**Reviewer #1:**
(1) The mechanism by which fenofibrate rescues memory loss in Kallistatin-transgenic mice is unclear. As a PPARalpha agonist, does fenofibrate target the Kallistatin pathway directly or indirectly? Please provide a discussion based on literature supporting either possibility.

Thank you for your important suggestion. Fenofibrate is indeed acting as a PPARα agonist. Fenofibrate has been shown to protect memory and cognitive function by downregulating α- and β-secretases[1]. Activation of PPARα can reduce Aβ plaques by upregulating ADAM10, thereby protecting memory and cognition[2]. Whereas, Fenofibrate can also act through a PPARα-independent pathway[3]. In our previous study, we proved that Fenofibrate can directly down-regulate the expression of Kallistatin in hepatocytes[4]. Here, our findings showed that Kallistatin induces cognitive memory deterioration by increasing amyloid-β plaques accumulation and tau protein hyperphosphorylation (Fig. 1-3), and Fenofibrate can directly down-regulate the serum level of Kallistatin (Fig. 8G). In addition, the expression of PPARα in the hippocampal tissue of Kallistatin (KAL-TG) mice showed no significant difference compared to the WT group (Author response image 1A-B). Therefore, we think Fenofibrate may improve memory and cognitive function at least in part through a PPARα-independent effect, which provides a new mechanism of Fenofibrate in AD with elevated Kallistatin levels.

**Author response image 1. sa4fig1:** Protein levels of PPARα were tested by western blot analysis in hippocampal tissue, then statistically analyzed the above results.

(2) The current study exclusively investigated the hippocampus. What about other cognitive memory-related regions, such as the prefrontal cortex? Including data from these regions or discussing the possibility of their involvement could provide a more comprehensive understanding of the role of Kallistatin in memory impairment.

Thank you for your suggestion. In addition to hippocampal tissue analysis, we performed immunohistochemical detection of Aβ and phosphorylated Tau levels in the prefrontal cortex. Our findings revealed that KAL-TG mice exhibited significantly elevated Aβ and phosphorylated Tau levels in the prefrontal cortex compared to WT mice. These observations align with the pathological patterns observed in hippocampal tissues, demonstrating consistent neurodegenerative pathology across both the hippocampus and prefrontal cortex. The data for this part are seen as follows.

**Author response image 2. sa4fig2:** Immunofluorescence staining of Aβ and phosphorylated tau (p-tau T231) was carried out in the prefrontal cortex tissue of KAL-TG and WT mice. Error bars represented the Standard Error of Mean (SEM); ***p* < 0.01. *Scale bar*, 100 μm.

(3) Fenofibrate rescued phenotypes in Kallistatin-transgenic mice while rosiglitazone, a PPARgamma agonist, did not. This result contradicts the manuscript's emphasis on a PPARgamma-associated mechanism. Please address this inconsistency.

Thank you for the reminder. In fact, our results showed a trend towards improved memory and cognitive function in KAL-TG mice treated with Rosiglitazone, although its effect is not as significant as that of Fenofibrate. Several studies have reported that Rosiglitazone has a beneficial effect on memory and cognitive function in mouse models of dementia, while these studies involve treatment periods of 3 to 4 months[5, 6], whereas our treatment period was only one month. Extending the treatment period with Rosiglitazone may result in a more pronounced improvement. In addition, Fenofibrate may have a PPAR-independent pathway by downregulating Kallistatin directly as discussed above and then show stronger effects.

(4) Most of the immunohistochemistry images are unclear. Inserts have similar magnification to the original representative images, making judgments difficult. Please provide larger inserts with higher resolution.

According to your suggestion, we provided larger inserts with higher resolution in Fig 3A and Fig 4B, as follows:

(5) The immunohistochemistry images in different figures were taken from different hippocampal subregions with different magnifications. Please maintain consistency, or explain why CA1, CA3, or DG was analyzed in each experiment.

Thank you for your advice. The trends of changes in different brain regions(including CA1, CA3, or DG) are consistent. Following your suggestion, we have now selected the DG region replaced the different hippocampal subregions with the DG area, and re-conducted the statistical analysis in Fig 5I & 6C, as follows. Due to the significant deposition of Aβ only in the CA1 region, Fig 2A was not replaced.

(6) Figure 5B is missing a title. Please add a title to maintain consistency with other graphs.

Thanks for your suggestion. We have added a title to Figure 5B, as follows:

(7) Please list statistical methods used in the figure legends, such as t-test or One-way ANOVA with post-hoc tests.

Thanks for your suggestion. We have listed the statistical methods used in the figure legends.

**Reviewer #2:**
(1) It was suggested that Kallistatin is primarily produced by the liver. The study demonstrates increased Kallistatin levels in the hippocampus tissue of AD mice. It would be valuable to clarify if Kallistatin is also increased in the liver of AD mice, providing a comprehensive understanding of its distribution in disease states.

Thank you for your suggestion. We extracted liver tissue from APP/PS1 mice, and the Western blot results indicated that the expression of Kallistatin in the liver of APP/PS1 mice was elevated, as follows:

**Author response image 3. sa4fig3:** Protein levels of Kallistatin were tested by western blot analysis in the liver tissue, then statistically analyzed the above results. Error bars represented the Standard Error of Mean (SEM); ***p* < 0.01.

(2) Does Kallistatin interact directly with Notch1 ligands? Clarifying this interaction mechanism would enhance understanding of how Kallistatin influences Notch1 signaling in AD pathology.

Thank you for your suggestion. This study reveals that Kallistatin directly binds to Notch1 and contributes to the activation of the Noch1-HES1 signaling pathway. As for whether Kallistatin can bind to the ligands of Notch1, it needs to conduct further investigations in future studies. Our preliminary data showed that Jagged1 was upregulated in the hippocampal tissues of KAL-TG mice by qPCR and Western blot analyses.

**Author response image 4. sa4fig4:** Kallistatin promoted Notch ligand Jagged1 expression to activate Notch1 signaling. (A) QPCR analysis of Notch ligands (Dll1, Dll3, Jagged1, Jagged2) expression in the 9 months hippocampus tissue. (B) Western blotting analysis of Notch ligand Jagged1 expression in the hippocampus tissue. (C) Western blotting analysis of Notch ligand Jagged1 expression in the hippocampus primary neuron. β-actin served as the loading control. Error bars represented the Standard Error of Mean (SEM); **p* < 0.05.

(3) Is there any observed difference in AD phenotype between male and female Kallistatin-transgenic (KAL-TG) mice? Including this information would address potential gender-specific effects on cognitive decline and pathology.

Thank you for your suggestion. Actually, we have previously used female mice for Morris Water Maze experiments, and the results showed that both male and female KAL-TG mice exhibited a phenotype of decreased memory and cognitive function compared to the gender-matched WT group, while there was no significant difference between male and female KAL-TG mice as follows:

**Author response image 5. sa4fig5:** Behavioral performance was assessed through the Morris water maze test. (A) The escape latency time was presented during 1-5 days. (B-D) Cognitive functions were evaluated by spatial probe test on day 6, then analyzing each group of mice crossing platform times(B), time percent in the targeted area (C), and the path traces heatmap (D). Error bars represented the Standard Error of Mean (SEM); F represents Female, M represents Male, and TG refers to KAL-TG; **p* < 0.05.

(4) It is recommended to include molecular size markers in Western blots for clarity and accuracy in protein size determination.

Thank you for your reminder. We have shown the molecular weight of each bolt.

(5) The language should be revised for enhanced readability and clarity, ensuring that complex scientific concepts are communicated effectively to a broader audience.

According to your suggestion, we have polished the article for enhancing readability and clarity.

**Reviewer #3:**
(1) The authors did not illustrate whether the protective effect of fenofibrate against AD depends on Kallistatin.

Thank you for your important suggestion. Fenofibrate is indeed acting as a PPARα agonist. Fenofibrate has been shown to protect memory and cognitive function by downregulating α- and β-secretases[1]. Activation of PPARα can reduce Aβ plaques by upregulating ADAM10, thereby protecting memory and cognition[2]. Whereas, Fenofibrate can also act through a PPARα-independent pathway[3]. In our previous study，we proved Fenofibrate can directly down-regulate the expression of KAL in hepatocytes[4]. Here, our findings showed that Kallistatin induces cognitive memory deterioration by increasing amyloid-β plaques accumulation and tau protein hyperphosphorylation (Fig. 1-3), and Fenofibrate can directly down-regulate the serum level of Kallistatin (Fig. 8G). In addition, the expression of PPARα in the hippocampal tissue of Kallistatin (KAL-TG) mice showed no significant difference compared to the WT group (Author response image 1-B). Therefore, we think Fenofibrate may improve memory and cognitive function at least in part through downregulatin Kallistatin. To conclusively determine whether fenofibrate’s therapeutic effects depend on Kallistatin, future studies should employ Kallistatin-knockout AD animal models to evaluate fenofibrate’s impact on cognitive and memory functions. These investigations will further clarify the mechanistic underpinnings of fenofibrate in AD therapy.

(2) The conclusions are supported by the results, but the quality of some results should be improved.

Thank you for your kind suggestion. We have updated the magnified images in the immunohistochemistry section of the article, ensuring that the fields of view for the immunohistochemistry are within the same brain region, and have shown the molecular weights in each bolt. Additionally, we have conducted a quantitative analysis of the protein levels in the Western blot results presented in Fig6&8.

(3) Figures 2c, 3c, and 4a present the Western blot results of p-tau from mice of different ages on one membrane, showing age-dependent expression. The authors analyzed the results of mice of different ages in one statistical chart, which will create ambiguity with the results of the representative images. For example, the expression of p-tau 396 in the blot was lower in the WT-12 M group than in the WT-9 M group (Figure 3c), which is contradictory to the statistical analysis.

Thank you for your reminder. The statistical presentation here does not match the figure. At that time, the WB experiments for the hippocampal tissue at each age group were conducted separately, and it was not appropriate to compare different age groups together. This graph cannot illustrate age dependency. We have replaced the statistical graph in Figure 3B&D, as follows:

(4) Figure 4b shows that KAL-TG-9 M had greater BACE1 expression than KAL-TG-12 M. Furthermore, the nuclei are not uniformly colored. Please provide more representative figures.

Thank you for your reminder. Due to the fact that these sets of data were not processed in a single batch, the ages in the graph are not comparable. Regarding the issue of inconsistent nuclear staining, we have provided another representative image from this group, as follows:

(5) Unclear why the BACE1 and Aβ levels seems less with KAL+shHES1 treatment than GFP+shNC treatment (Fig 6H)? This finding contradicts the conclusion.

Thank you for your reminder. This experiment was repeated three times, and here, we have represented the representative results along with the corresponding statistical data. There are no difference between KAL+shHES1 treatment and GFP+shNC treatment. We have updated the Fig. 6H.

(6) The Western blot results in figure 6e-h, 8h-i, and S3-S5 were not quantified.

Thank you for your reminder. We have added statistical graphs and original images of the pictures in figure 6e-h, 8h-i, and S3-S5.

(7) The authors did not provide the detection range of the Aβ42 ELISA kit.

Thank you for your suggestion. The Aβ42 ELISA kit is from the IBL, with the product number 27721. Its standard range is 1.56 - 100 pg/mL, and the sensitivity is 0.05 pg/mL.

(8)The authors did not specify the sex of the mice. This is important since sex could have had a dramatic impact on the results.

Thank you for your suggestion. The results we present in the text are all statistically obtained from male mice. Actually, we have previously used female mice for Morris Water Maze experiments, and the results showed that both male and female KAL-TG mice exhibited a phenotype of decreased memory and cognitive function compared to the gender-matched WT group, while there was no significant difference between male and female KAL-TG mice (Author response image 5).

Minor:(1) In Figure 2b, there are no units for the vertical coordinates of the statistical graph.

Thank you for your reminder. We have added units for the vertical coordinates in Figure 2b.

(2) In Figure 2c, the left Y-axis title is lacking in the statistic chart.

Thank you for your reminder. We have added the left Y-axis title in the statistic chart.

Reference：

(1) Assaf N, El-Shamarka ME, Salem NA, Khadrawy YA, El Sayed NS. Neuroprotective effect of PPAR alpha and gamma agonists in a mouse model of amyloidogenesis through modulation of the Wnt/beta catenin pathway via targeting alpha- and beta-secretases. Progress in Neuro-Psychopharmacology and Biological Psychiatry 2020, 97: 109793.

(2) Rangasamy SB, Jana M, Dasarathi S, Kundu M, Pahan K. Treadmill workout activates PPARα in the hippocampus to upregulate ADAM10, decrease plaques and improve cognitive functions in 5XFAD mouse model of Alzheimer’s disease. Brain, Behavior, and Immunity 2023, 109: 204-218.

(3) Yuan J, Tan JTM, Rajamani K, Solly EL, King EJ, Lecce L, et al. Fenofibrate Rescues Diabetes-Related Impairment of Ischemia-Mediated Angiogenesis by PPARα-Independent Modulation of Thioredoxin-Interacting Protein. Diabetes 2019, 68(5): 1040-1053.

(4) Fang Z, Shen G, Wang Y, Hong F, Tang X, Zeng Y, et al. Elevated Kallistatin promotes the occurrence and progression of non-alcoholic fatty liver disease. Signal Transduct Target Ther 2024, 9(1): 66.

(5) Nelson ML, Pfeifer JA, Hickey JP, Collins AE, Kalisch BE. Exploring Rosiglitazone's Potential to Treat Alzheimer's Disease through the Modulation of Brain-Derived Neurotrophic Factor. Biology (Basel) 2023, 12(7).

(6) Pedersen WA, McMillan PJ, Kulstad JJ, Leverenz JB, Craft S, Haynatzki GR. Rosiglitazone attenuates learning and memory deficits in Tg2576 Alzheimer mice. Exp Neurol 2006, 199(2): 265-273.